

# Moderate Greenland ice sheet melt during the last interglacial constrained by present-day observations and paleo ice core reconstructions

P. M. Langebroek[1] and K. H. Nisancioglu[2,3]

[1]Uni Research Climate, Bjerknes Centre for Climate Research, Allégaten 55, 5007 Bergen, Norway
[2]Department of Earth Science, University of Bergen and Bjerknes Centre for Climate Research, Allégaten 70, 5007 Bergen, Norway
[3]Centre for Earth Evolution and Dynamics, Postbox 1028 Blindern, University of Oslo, 0315 Oslo, Norway

*Correspondence to:* P. M. Langebroek (petra.langebroek@uni.no)

**Abstract.**

During the last interglacial period (LIG, ~ 130–115 ka before present, ka = 1000 yr) summer temperatures over Greenland were several degrees higher than today. It is likely that the Greenland ice sheet (GIS) was smaller than today, contributing to the reconstructed sea-level highstand of the LIG. However, the range of simulated GIS melt is large, and the location of the melt

is uncertain. Here, we use temperature and precipitation patterns simulated by the Norwegian Earth System Model (NorESM) to investigate the volume, extent and stability of the GIS during the LIG. Present-day observations of ice sheet size, elevation and stability, together with paleo elevation information from five deep ice cores, are used to evaluate our ensemble of GIS simulations. Accepted simulations indicate a maximum GIS reduction equivalent to a global mean sea-level rise of 0.8–2.2 m compared to today, with most of the melt occurring in the southwest. The timing of the maximum ice melt over Greenland is

simulated between 124 and 122 ka.

We furthermore suggest a preferred mean value for the basal sliding parameter, relatively high PDD factors and an average to high atmospheric temperature lapse rate based on training the SICOPOLIS ice sheet model to observations and available LIG proxy data.

## 1 Introduction

In order to better understand the impact of future climate warming on the stability of the Greenland ice sheet (GIS), it is valuable to investigate its behaviour during past warm periods. One of the most recent periods of warming is the last interglacial period (LIG, ~ 130–115 ka before present, ka = 1000 yr). During this period annual mean temperatures were similar to today. However, due to a different orbital configuration, the seasonal cycle was stronger and boreal summers were several degrees warmer during the early LIG compared to today (e.g. Masson-Delmotte et al., 2013; Langebroek and Nisancioglu, 2014).

For the Arctic, reconstructions indicate a peak warming of up to 8 °C (CAPE Last Interglacial Project Members, 2006). Over central and east Greenland summer temperatures were likely 4-5 °C higher than present and on the coast approximately 3 °C above present (CAPE Last Interglacial Project Members, 2006; Otto-Bliesner et al., 2006; Alley et al., 2010; Axford et al.,





2011). Eustatic sea level was $\sim 6$ to $9\,\mathrm{m}$ above present (Kopp et al., 2009; Dutton and Lambeck, 2012). Thermal expansion of the ocean and melting of mountain glaciers caused approximately $1\,\mathrm{m}$ of the total eustatic sea-level rise (Dutton et al., 2015). The remaining 5 to $8\,\mathrm{m}$ sea-level rise was due to the melting of the polar ice sheets. To what extent the GIS contributed to this sea-level highstand is uncertain, but it is likely to have accounted for approximately $2\,\mathrm{m}$ (Kopp et al., 2009; Colville et al.,

5 2011).

Reconstructing the extent of the LIG GIS is difficult, as later glaciations eroded most of the direct evidence of its margins. The only data available comes from the few deep ice cores (e.g. NGRIP-members, 2004; NEEM community members, 2013), or indirectly by assessing sediment discharge from Greenland (e.g. Colville et al., 2011). In all deep ice cores (Camp Century, NEEM, NGRIP, GRIP and Dye3) LIG ice is found (e.g. NGRIP-members, 2004; NEEM community members, 2013), suggest-

ing a relatively large GIS survived the LIG. However, interpretation of the basal part of the ice cores is not straight forward, and the existence of LIG ice is discussed, especially for Dye3. Some studies claim that Dye3 was ice free during the LIG (Koerner, 1989; Koerner and Fischer, 2002), indicating a much smaller remaining GIS. However, analyses of biological material suggest that ice was present at Dye3 over the past 400ka (Willerslev et al., 2007). Alley et al. (2010) disagree, suggesting instead that this material could also have survived the LIG deglaciation by being preserved in permafrost instead of in ice. Nonetheless,

indirect evidence from isotope ratios in sediment discharged from Greenland support only a minor deglaciation of the southern part of the GIS, suggesting that Dye3 was covered by ice (Colville et al., 2011). Recently, a new technique of mapping ice horizons by ice-penetrating radar surveys finds no LIG ice in southern Greenland (MacGregor et al., 2015). However, this technique also predicts absence of LIG ice at Camp Century, GRIP and NGRIP, indicating that the radar might not detect small amounts of (LIG) ice.

In the past there have been numerous endeavours at simulating the LIG GIS (Fig. 1), resulting in a large range of possible scenarios for the amount and source of ice loss from Greenland. The simulated GIS contribution to LIG sea-level rise varies between 0.3 and $5.9\,\mathrm{m}$ (see Fig. 1, and references therein). In all studies central Greenland remained covered by ice, and the largest melting occurred either in the north (Fyke et al., 2011; Quiquet et al., 2013; Stone et al., 2013), the southwest (Letréguilly et al., 1991; Greve et al., 1998; Tarasov and Peltier, 2003; Otto-Bliesner et al., 2006), or both (Ritz et al., 1997;

Cuffey, 2000; Huybrechts, 2002; Lhomme et al., 2005; Robinson et al., 2011; Born and Nisancioglu, 2012; Helsen et al., 2013; Calov et al., 2015). Probably the main reason for the wide range in results is the difference in climate forcing and method of coupling of this forcing to the ice sheet model, resulting in a different surface mass balance for Greenland. Other differences, such as initial bedrock and ice topographies, geothermal heat flux, and ice sheet model physics, may also play a role in the simulated spread.

The earlier studies assume that the present-day distribution of temperature stays constant and is only varied by a spatially uniform climate forcing, i.e. the 'index' method. The climate, or temperature, forcing is derived from a $\delta^{18}\mathrm{O}$ record retrieved from an ice core. In most studies, the GRIP ice core record (Dansgaard et al., 1993) is used as a reference, sometimes extended further back in time by using the Vostok record (Petit et al., 1999). In later studies the surface mass balance is derived from simulations by general circulation models (GCM) forced with LIG boundary conditions (orbital configuration and greenhouse

gases). In most of these studies the climate model forcing is applied off-line, meaning that changes in ice topography and



albedo do not feed back to the GCMs. This is because GCM simulations are computationally expensive, so only snap-shot or equilibrium simulations are performed for the LIG. Between the different equilibrium states, simulated temperature and precipitation fields are interpolated over the 20 kyr duration of the LIG. Some studies use intermediate complexity climate models (Robinson et al., 2011; Calov et al., 2015) or an atmosphere-only GCM (Helsen et al., 2013) in combination with a

regional climate model (RCM) to compute a higher resolution transient climate forcing for the ice sheet model.

In many of the studies, an ensemble of model simulations is performed, giving a range of possible LIG GIS melt scenarios (e.g. Robinson et al., 2011; Born and Nisancioglu, 2012). Based on constraints defined by geological data, unlikely model simulations are discarded. In the past, the existence of LIG ice on Dye3 has been used as criteria to accept or discard model results. For example, Otto-Bliesner et al. (2006) estimated the minimum melting of the GIS from the moment Dye3 becomes

ice-free. According to the latest research it is likely that LIG persisted at Dye3, therefore Otto-Bliesner et al. (2006) probably rejected scenarios that now would be considered realistic. Thus, the interpretation of the geological data strongly affects the accepted model outcome.

In this study, we use the thermomechanically coupled ice sheet model SICOPOLIS (Greve, 1997a, b) to reconstruct the LIG GIS, and assess the GIS contribution to the observed LIG sea-level highstand. The ice model is forced with temperature and

precipitation patterns extracted from four LIG equilibrium simulations with the Norwegian Earth System Model (NorESM). Constraints from present-day observations and paleo reconstructions are applied in order to find a likely range of LIG GIS melt.

In Sec. 2 the essentials of the climate and ice sheet models are described, followed by an explanation of the model simulation set-up, a description of the model parameters tested, and a list of present-day and paleo constraints used in this study. Section 3

first discusses the temperature and precipitation patterns as simulated by the NorESM, followed by a description of the three different simulated states of the GIS: the modern GIS (control simulations), the glacial GIS (spin-up or initialization for LIG simulation), and the LIG GIS. The results are discussed in light of previous work in Sect. 4, followed by suggestions for further research. The conclusions are summarized in Sect. 5.

## 2   Methods and experimental set-up

The climate fields necessary to force the SICOPOLIS ice sheet model are computed using the Norwegian Earth System Model (NorESM). Unfortunately, it is too computationally expensive to fully couple the ice sheet and climate models for the period of interest (LIG, ∼20 kyr). Instead, we use an uncoupled approach where time-slice experiments with NorESM are interpolated in time and space and applied as forcing for the ice sheet model. Here we shortly describe the two numerical models, followed by a detailed description of the experimental set-up, and a list of the modern and paleo data used to constrain the models.

Possible shortcomings introduced by running the models un-coupled are addressed in the discussion section.





## 2.1 The climate model

The Norwegian Earth System Model (NorESM) is derived from the Community Earth System Model (CESM) developed at the National Centre for Atmospheric Research (NCAR). It consists of the same components for the atmosphere (CAM4), land (CLM4) and sea ice (CICE4), and uses the CESM coupler (CLP7). However, a key difference with CESM is the use of a different ocean component, based on the Miami Isopycnic Coordinate Ocean Model (MICOM). This component is largely modified from MICOM in order to improve conservation of mass and heat, and the efficiency and robustness of the transport of tracers (for more details see Assmann et al., 2010). NorESM is one of the models used in the IPCC assessments reports and is the Norwegian contribution to the Climate Model Intercomparison Project (CMIP; e.g. Taylor et al., 2012). For a detailed description of NorESM and its simulated present-day and future climate states we refer to Bentsen et al. (2013) and Iversen et al. (2013).

For this study the simulations are performed with the low-resolution version of NorESM (NorESM-L) in order to reduce computational time for the relatively long equilibrium simulations. The atmospheric component has a spatial resolution of approximately $3.75° \times 3.75°$ (T31) and comprises 26 levels in the vertical. The ocean component's horizontal grid size corresponds to a nominal grid size of $3°$ (g37) and consists of 30 isopycnic layers in the vertical. The sea ice component follows the ocean grid, and the land component follows the atmospheric grid. For further details concerning the different components within NorESM-L and a thorough description of the pre-industrial results, we refer to Zhang et al. (2012).

## 2.2 The ice sheet model

We use the SICOPOLIS ice sheet model (version 3.1, Greve, 1997a, b) in order to simulate the GIS. SICOPOLIS is a three-dimensional thermomechanical ice sheet model, using the shallow-ice approximation (SIA) to solve the evolution of ice thickness over time. This approximation neglects longitudinal stresses in the ice, and is therefore only valid for ice masses that are thin compared to their horizontal extent. The advantage of using the SIA is that it is computationally much more efficient than solving higher orders of the dynamics stress balance as is done in higher order or full-Stokes ice flow models. It is a valid, and widely used, approach for the GIS. However, the margins of the GIS with fast flowing ice (e.g. outlet glaciers) cannot accurately be simulated using the SIA. This generally results into a mismatch between simulated and observed margins of the GIS, where ice sheet models using the SIA often overestimate the thickness and extent of the ice (e.g. Greve et al., 2011; Robinson et al., 2011; Quiquet et al., 2013; Stone et al., 2013; Dolan et al., 2015). It is not possible to model ice shelves using the SIA. However, as we focus on a GIS that is similar to or smaller then its modern equivalent, ice shelves do not play a significant role and the SIA is a valid simplification. For a full description of the SICOPOLIS model we refer to Greve (1997a).

We use SICOPOLIS with its standard grid resolution for Greenland, which consists of 90 vertical layers and a horizontal resolution of $20\,\mathrm{km}$. Modern and glacial simulations are initiated using modern ice and bedrock topographies from Bamber et al. (2013). In contrast, the LIG simulations start from an equilibrium glacial representation of the GIS (see Sect. 3.3 and 3.4). The solid earth below the ice responds to changes in ice load by adjusting the bedrock topography assuming an elastic



lithosphere and relaxing asthenosphere, the so-called ELRA model. See Le Meur and Huybrechts (1996) for an overview of the different isostasy models.

The geothermal heat flux is assumed to be constant over time. It is based on the global heat flow representation of Pollack et al. (1993), which is modified by Greve (2005) in order to fit measured basal temperatures at four ice core locations (GRIP,

NGRIP, Camp Century and Dye3) and observed present-day ice thickness. The applied heat-flux map shows increasing values from west to east, with a relatively high heat-flux anomaly in the central north (around NGRIP) and a low heat-flux anomaly in the south (around Dye3). It is likely that the spatial variations are larger (see also Greve, 2005) and vary over time. However, more detailed information is not available, and a thorough assessment of the effect of heat-flux variations on the GIS size, extent and stability is beyond the scope of this study.

The flow enhancement factor is kept fixed at the default value of 3. However, three different values for the basal sliding parameter are assessed in the sensitivity experiments (see Sect. 2.3.3).

Climate forcing is provided by the NorESM-L simulations. The simulated fields of temperature and precipitation are bilaterally interpolated to match the higher horizontal resolution of the ice sheet model. Temperatures are furthermore corrected for the difference in surface topography between the climate and ice sheet models by using a fixed atmospheric lapse rate.

Assuming a spatially and temporally constant lapse rate for Greenland is not realistic, but unfortunately no dataset exists describing how the atmospheric lapse rate varies over time and space. In order to assess the effect of a different lapse rate on the simulated GIS, we investigate three different values for the spatially uniform lapse rate (see Sect. 2.3.3). Within SICOPOLIS, the elevation corrected annual and July mean temperatures are converted to monthly values using a sine function. Hereafter, the temperatures are used to calculate melt following the positive degree day (PDD) method (Reeh, 1991). This assumes that

the amount of surface melt occurring is proportional to the sum of the temperatures on days above freezing. The two factors relating temperature to the amount of ice and snow lost per day are uncertain; therefore a range of values is tested (see Sect. 2.3.3).

Whether the precipitation from the climate model falls as snow or rain depends on the monthly temperatures. All precipitation is converted to snow if the temperature is -10 °C or below. Above 7 °C the precipitation is considered to be rain. In between

these temperatures a linear interpolation gives the relative amount of snow and rainfall.

## 2.3  Experimental set-up

### 2.3.1  Climate model simulations

We performed five time-slice simulations with NorESM-L: one pre-industrial (PI) and four LIG simulations (see Table 1, and see also Langebroek and Nisancioglu, 2014). All simulations use modern land-sea distribution, topography/bathymetry,

vegetation and ice sheet configuration.

The PI control simulation is run for 1000 yr under constant PI GHG levels and an orbital configuration fixed to the year 1950 (Berger, 1978). The LIG simulations represent four time slices each 5000 yr apart (130, 125, 120 and 115 ka) with orbital forcing set accordingly. The GHG levels of the oldest two simulations (130 and 125 ka) are set following the guidlines of the





Paleoclimate Modelling Intercomparison Project 3 (PMIP3) and are based on ice core data (e.g. Petit et al., 1999; Lüthi et al., 2008; Loulergue et al., 2008). For the younger LIG simulations (120 and 115 ka) ice core data indicates near PI GHG values, and therefore we keep the GHG forcing at PI levels. All LIG simulations are branched off from the PI simulation at model year 495, when the PI simulation is close to equilibrium, and then run for another 505 yr with updated orbital settings and

GHG values. The climate model results used in this study are based on the long-term mean values of years 900-1000 of each simulation, at which state the simulations are close to equilibrium. The experimental setup and results of the climate model are described in detail in Langebroek and Nisancioglu (2014). The results relevant for this study are discussed in Sect. 3.1.

### 2.3.2  Ice sheet model simulations

We simulate the GIS for three time periods: 1) PI or modern, 2) glacial preceding the LIG, and 3) LIG.

PI simulations are performed in order to compare the modelled modern ice sheet size, surface elevation and surface mass balance to long-term mean observations (see Sect. 3.2). All PI simulations are run for 100 kyr, starting from modern ice and bedrock elevations (after Bamber et al., 2013). The NorESM-L PI simulation provides the control long-term mean climate forcing, which is fixed over time. The interpolated ice sheet temperatures, however, do vary over time as they are adjusted with respect to the surface elevation difference between the two models. Topographies are fixed in NorESM-L, but vary in the ice

sheet model, when the ice sheet grows or melts. The GIS reaches equilibrium with the climate forcing within 30–40 kyr. The final equilibrium state after 100 kyr is used for the analyses (Sect. 3.2).

    Simulations representing the glacial state preceeding the LIG (∼135 ka, hereafter called preLIG) are included in order to provide a glacial initial state for the LIG simulations. preLIG simulations are initialized with modern ice and bedrock elevations, and are run for 100 kyr. Due to the lack of an appropriate glacial climate simulation the ice model is forced with fixed PI

NorESM-L climate fields, where temperatures over Greenland are reduced by 10 °C . This is an estimate based on the difference in $\delta^{18}$O found in the Vostok ice core between 135 ka and present (∼4 ‰) and a $\Delta$T/$\delta^{18}$O conversion factor of ∼2.4 °C / ‰ (see Huybrechts, 2002). This is consistent with the 8–12 °C glacial-interglacial temperature difference proposed by Petit et al. (1999) for Antarctica, but smaller than the likely Greenland temperature difference between the Last Glacial Maximum and present from borehole thermometry (approximately 20 °C, e.g. Johnsen et al., 2001). The NorESM-L precipitation field is not

modified for the preLIG simulations. However the relative amount of snow and rainfall is affected as this depends on the monthly temperature. The unchanged (PI) precipitation together with the colder (preLIG) temperatures, result in a relatively high snowfall. This contradicts reconstructions showing less snowfall during cold periods at high surface elevations (e.g. Dahl-Jensen et al., 1993; Cuffey and Clow, 1997). Unfortunately, as we do not have any constraints on the amount of snow accumulation during this glacial period from either data reconstructions (most Greenland ice cores do not reach that far back

in time) or a representative NorESM-L simulation, we have no means to better quantify the snow accumulation. The resulting preLIG ice sheet simulations reach equilibrium within 70–80 kyr and are discussed in Sect. 3.3.

    A suite of LIG simulations is performed in order to quantify the minimum GIS volume and extent. In contrast to the PI and preLIG simulations with fixed climate forcing, the LIG simulations are run for 20 kyr with transient climate forcing for the period 135–115 ka. The climate forcing is extracted from the four NorESM-L LIG simulations (at 130, 125, 120 and 115 ka).



To initialize the run, the ice sheet simulations all start at 135 ka with preLIG glacial ice and bedrock elevations as well as preLIG climate forcing (NorESM-L PI minus 10 °C). The climate forcing is spline-wise interpolated for every time step and grid cell of the ice sheet model between these five forcing time slices.

### 2.3.3 Perturbed model parameters

As noted in Sect. 2.2, there are several unconstrained parameters in the ice sheet model. The parameters evaluated in this study are 1) the basal sliding parameter, 2) the atmospheric temperature lapse rate and 3) the PDD factors for snow and ice.

1) The **basal sliding parameter** defines the ice velocity at the base of the ice sheet for regions where the ice is not frozen to the bedrock. A higher basal sliding parameter increases the basal ice velocities, thereby decreasing the surface slope of the ice and decreasing its total volume. It is unlikely that the basal sliding parameter is spatially uniform, as it depends on the basal material (sediments/bedrock) the ice flows over. However, as the distribution of sediments is poorly known for the present and paleo GIS, we consider a constant basal sliding parameter. The default SICOPOLIS value is $11.2\,\mathrm{m\,a^{-1}Pa^{-1}}$. We furthermore examine the effect of 5 and $17\,\mathrm{m\,a^{-1}Pa^{-1}}$ on the PI GIS.

2) The **atmospheric temperature lapse rate** is needed to correct the NorESM-L temperatures for the difference in surface topography between the climate and ice sheet model. The surface topography over Greenland is fixed in NorESM-L, whereas it evolves in SICOPOLIS. Due to lack of spatial and temporal data, we apply a constant, uniform temperature lapse rate. As default we use the standard mean atmospheric temperature lapse rate of $6.5\,\mathrm{°C\,km^{-1}}$. The effect of a particular lapse rate on the GIS size and extent is assessed by testing lapse rates of 5.0 and $8.0\,\mathrm{°C\,km^{-1}}$, representing wetter and drier air, respectively.

3) The **PDD factors for snow and ice** are used to calculate the amount of surface melt occurring, following the PDD method of Reeh (1991). The amount of melt is proportional to the sum of the temperatures on days above freezing. Higher PDD factors therefore cause more melting resulting into a smaller ice sheet. Lower PDD factors generate less surface melting and a more stable and larger ice sheet. Observations at different glaciers all over the world give factors varying between $3$–$6\,\mathrm{mm\,water\,day^{-1}\,°C^{-1}}$ for snow and $5$–$14\,\mathrm{mm\,water\,day^{-1}\,°C^{-1}}$ for ice (see Table 2 in Braithwaite, 1995). The PDD factors are not only location dependent, but likely also vary with season (e.g. Braithwaite and Olesen, 1993). Unfortunately no detailed maps of PDD factors exist, and we assume the factors to be constant. The only exception is SICOPOLIS' default PDD factor for ice. This depends on summer temperature and latitude. North of $72\,°N$ the PDD factor for ice is temperature dependent, using $15\,\mathrm{mm\,water\,day^{-1}\,°C^{-1}}$ for low summer temperatures ($\leq$ -1 °C) and $7\,\mathrm{mm\,water\,day^{-1}\,°C^{-1}}$ for high summer temperatures ($\geq$ +10 °C). An interpolation is applied for summer temperatures inbetween. South of $72\,°N$ the PDD factor for ice is temperature independent and set to $7\,\mathrm{mm\,water\,day^{-1}\,°C^{-1}}$. For simplicity we do not make this separation while testing the sensitivity of the PDD factor for ice, and rather apply a constant value (7, 10, 15 or $20\,\mathrm{mm\,water\,day^{-1}\,°C^{-1}}$). The default PDD factor for snow is independent of temperature and latitude, and set to $3\,\mathrm{mm\,water\,day^{-1}\,°C^{-1}}$. We also assess values of 5 and $8\,\mathrm{mm\,water\,day^{-1}\,°C^{-1}}$. We tested the entire set of 12 combinations of ice and snow PDD factors in the PI, preLIG and LIG simulations. However, the PDD factor of snow should be smaller than the PDD factor of ice (Braithwaite, 1995). Our combination of PDD factors 8 and $7\,\mathrm{mm\,water\,day^{-1}\,°C^{-1}}$ for snow and ice, respectively, is therefore not realistic, but is just shown for sake of completeness.



The range of parameters examined in our simulations (see Table 2) is similar to the range selected by other recent studies of the LIG GIS (e.g. Robinson et al., 2011; Applegate et al., 2012; Born and Nisancioglu, 2012; Stone et al., 2013). By comparing the simulations with observations and reconstructions of the present-day and LIG GIS we attempt to reduce the likely range of combinations of these parameters, and simultaneously obtain the mostly likely size of the minimum LIG GIS.

## 2.4 Modern and paleo constraints

To assess the PI control simulation of the GIS we use the latest estimate of the present-day GIS volume (∼7.4 m sea-level equivalents, SLE) and ice surface elevation (Bamber et al., 2013).

We also assess the ratio between surface mass balance (SMB) and total accumulation of our PI simulations, following Robinson et al. (2011) and Calov et al. (2015). This ratio provides a measure for the stability of the GIS. High ratios indicate little surface melt and therefore a stable ice sheet. In contrast, low ratios indicate more surface melt and an unstable ice sheet. The ratio is thought to be approximately 50% for the present-day ice sheet (e.g. Ettema et al., 2009; Alley et al., 2010; Applegate et al., 2012), but a range of 45–65% is accepted in this study (similar to Robinson et al., 2011; Calov et al., 2015).

Furthermore, for the LIG ice sheet simulations we assume that ice was present on all deep ice core locations during the LIG, including Dye3. And we consider only a minor reduction in surface elevation at central ice core locations compared to present (∼100 m, Raynaud et al., 1997).

## 3 Results

### 3.1 Simulated temperature and precipitation

In Langebroek and Nisancioglu (2014) we investigated the timing of peak warmth during the LIG as computed by the NorESM-L simulations (Sect. 2.3.1). We focused on understanding the LIG sea-surface temperature (SST) evolution in the North Atlantic, and compared the modelled SSTs to reconstructed SSTs from sediment cores. We found that in general the simulations captured the North Atlantic SST trend well. However, NorESM-L underestimated the absolute SSTs in the northern part of the North Atlantic by ∼2–3 °C (Langebroek and Nisancioglu, 2014). In contrast, for the two locations just south of Greenland, modelled SSTs are very similar to reconstructed values. The most likely cause for the northern North Atlantic SST cold bias in the model is an underestimation of inflow of warm Atlantic water into the Nordic Seas. This might also result in relatively cold surface air temperatures over northern Greenland.

For most of Greenland the simulated PI annual and July mean temperatures (Tann and Tjul, respectively) are below zero (Fig. 2). Only the southern tip of Greenland exhibits temperatures slightly above freezing. The lowest temperatures are located in the inland part of Greenland, central for Tjul, and more to the north for Tann. Comparison of the simulated temperature patterns to ERA40 reanalysis data shows that NorESM-L is on average a few degrees too cold. The largest cold bias is on the coast, whereas NorESM-L is too warm over central Greenland. The latter is due to the relatively low Greenland surface topography in NorESM-L of up to about 1 km lower than observed for central Greenland (Fig. 3). However, this misfit will





not affect the ice sheet model simulations much. Surface temperatures fed into the ice sheet model are corrected for this surface elevation difference, and are therefore in general colder than simulated by NorESM-L. The surface topography in the ice sheet model evolves over time, as it equilibrates to the climate forcing. The final temperature patterns are therefore strongly simulation dependent, and can only be discussed in the framework of the specific ice sheet model simulations (see next sections).

The NorESM-L simulated PI precipitation pattern shows relatively dry conditions in the central north (Fig. 2). Precipitation increases towards the south, with maximum precipitation found at the southeast coast. This pattern is also found in other climate model simulations (e.g. Born and Nisancioglu, 2012) and in the ERA40 reanalysis dataset. The NorESM-L PI simulation is, however, overall dryer than ERA40. It especially underestimates the extreme increase in precipitation at the southeastern flank of Greenland as found by ERA40. The low resolution of NorESM-L cannot realistically capture the mountain slopes in this region (Fig. 3) and therefore the model simulates a much smoother precipitation gradient than is observed (Fig. 2).

For the LIG, NorESM-L simulates Tann patterns relatively similar to the PI (Fig. 4). Moderate warming (cooling), up to $2\,^{\circ}\mathrm{C}$, is found at the northernmost tip of Greenland at $125\,\mathrm{ka}$ ($115\,\mathrm{ka}$) (Fig. 4, upper row). Larger temperature deviations are simulated for Tjul, where NorESM-L shows a temperature increase everywhere on Greenland up to approximately $4\,^{\circ}\mathrm{C}$ during the early LIG (130 and $125\,\mathrm{ka}$). This is similar to, but slightly less extreme than, the reconstructed LIG summer temperatures of $4$–$5\,^{\circ}\mathrm{C}$ above present over central and eastern Greenland and the approximately $3\,^{\circ}\mathrm{C}$ above present on the coast (CAPE Last Interglacial Project Members, 2006; Otto-Bliesner et al., 2006; Alley et al., 2010; Axford et al., 2011). In contrast, during the late LIG ($115\,\mathrm{ka}$) Greenland Tjul are up to $4\,^{\circ}\mathrm{C}$ lower than during the PI. The strong seasonal amplification and rather weak annual mean variations are the results of the orbital forcing, which has been shown to have a strong impact on LIG climate (e.g. Langebroek and Nisancioglu, 2014).

The mean LIG precipitation over Greenland as simulated by NorESM-L does not deviate much from the PI (see mean difference values in Fig. 4). However, locally large differences occur. Precipitation in the generally wet southeast Greenland is reduced by up to $30\,\mathrm{mm\,yr^{-1}}$ in the early LIG (130 and $125\,\mathrm{ka}$). In contrast a similar increase is found during the late LIG (120 and $115\,\mathrm{ka}$). Depending on the exact location these changes account for $\sim 5$ to $20\%$ (south to east respectively) of the annual mean precipitation. The opposite is simulated for the northwest corner of Greenland, with an increase (decrease) in precipitation of up to approximately $20\%$ at 130 and $125\,\mathrm{ka}$ (120 and $115\,\mathrm{ka}$).

From these temperature and precipitation patterns a general trend of early LIG ice sheet melting and late LIG ice sheet growth is to be expected.

### 3.2 Control: Modern Greenland ice sheet

When applying the LIG climate forcing described above and allowing for the entire spectrum of possible values for the uncertain parameters (basal sliding, atmospheric temperature lapse rate and PDD factors for snow and ice), the simulated LIG minimum GIS ranges from hardly any reduction to an almost entirely vanished ice sheet. Not all of these solutions, however, have a realistic present-day counterpart. In this section we therefore first assess the modern GIS as simulated by the PI NorESM-L run.





Figure 5 shows the total Greenland ice volume as simulated for different values of the basal sliding parameter (columns), sets of PDD factors (coloured dots) and atmospheric temperature lapse rate (horizontal bars). The majority of the solutions have an ice volume similar to the observed value of Bamber et al. (2013, $\sim 7.4\,\mathrm{m}\,\mathrm{SLE}$). Lower (higher) basal sliding causes the shape of the ice sheet to change and results in a larger (smaller) ice volume. Changing the atmospheric lapse rate (between

$5, 6.5$ and $8\,^{\circ}\mathrm{C}\,\mathrm{km}^{-1}$) only has a small effect on the total ice volume ($\pm \sim 0.1\,\mathrm{m}\,\mathrm{SLE}$). In contrast, different PDD factors affect the resulting ice volume significantly. For the highest basal sliding scenario ($17\,\mathrm{m}\,\mathrm{a}^{-1}\,\mathrm{Pa}^{-1}$) the highest PDD factors (8 and $20\,\mathrm{mm}\,\mathrm{water}\,\mathrm{day}^{-1}\,^{\circ}\mathrm{C}^{-1}$ for snow and ice, respectively) gives an ice sheet more than 15% (or $\sim 1.1\,\mathrm{m}\,\mathrm{SLE}$) smaller than observed. Not including this simulation, the range in PDD factors still generates a spread of approximately $0.8\,\mathrm{m}\,\mathrm{SLE}$.

Even though the total ice volume is close to or even slightly larger than observed for most of the simulations, the simulated

surface elevation for large parts of inland Greenland is lower then observed (Fig. 6). At the central Greenland deep ice core locations (NEEM, NGRIP and GRIP) simulated ice surface elevations are slightly below observed. A larger mismatch is found at Dye3 (Fig. 6c). Here all simulations using the upper extreme of basal sliding ($17\,\mathrm{m}\,\mathrm{a}^{-1}\,\mathrm{Pa}^{-1}$) give a surface elevation more than 15% lower then observed. In the remainder of this study we therefore omit simulations using the basal sliding parameter of $17\,\mathrm{m}\,\mathrm{a}^{-1}\,\mathrm{Pa}^{-1}$.

In general, the simulated ice sheet extents further towards the coast than observed, especially in the northern and eastern parts of Greenland, but also in the southwest. Due to the excessive ice in the coastal areas, ice surface elevations at Camp Century and the future drill site EGRIP are overestimated. Many SIA ice sheet models simulate a too flat modern GIS, with too high elevations on the coast (e.g. Stone et al., 2010, 2013; Applegate et al., 2012). Even the hybrid model of Quiquet et al. (2013), that includes the Shallow Shelf Approximation in order to better resolve the faster flowing ice streams near the coast,

overestimates coastal ice elevation by up to $1000\,\mathrm{m}$. In a recent study, Calov et al. (2015) manage to simulate the ice sheet edge much closer to its observed position, by including a new discharge parameterization. Including this kind of parameterizations can improve future simulations of the GIS.

The default solution for basal sliding ($11.2\,\mathrm{m}\,\mathrm{a}^{-1}\,\mathrm{Pa}^{-1}$) gives ice volumes closest to the observed value (Fig. 5a). Also the root mean square errors of simulated ice surface elevations compared to observations (Bamber et al., 2013) are smallest

for this subset of the ensemble. For simplicity, we consider $11.2\,\mathrm{m}\,\mathrm{a}^{-1}\,\mathrm{Pa}^{-1}$ our best basal sliding solution and use this for further investigation. However we have to keep in mind that the upper range of the PDD factors (5 & $8\,\mathrm{mm}\,\mathrm{water}\,\mathrm{day}^{-1}\,^{\circ}\mathrm{C}^{-1}$ for snow and $20\,\mathrm{mm}\,\mathrm{water}\,\mathrm{day}^{-1}\,^{\circ}\mathrm{C}^{-1}$ for ice) causes a too low surface elevation at Dye3 (more than 15% below observed) (Fig. 6c).

In order to assess the stability of the simulated GIS, we investigate the ratio of SMB versus total accumulation. Previous

studies suggest that the modern ratio is 45-65% for Greenland (e.g. Ettema et al., 2009; Alley et al., 2010; Robinson et al., 2011; Applegate et al., 2012; Calov et al., 2015). PDD factors define the amount of surface melt and therefore strongly affect this ratio. Higher (lower) PDD factors cause more (less) melt and a less (more) stable GIS. More than half of the simulations with a default basal sliding ($11.2\,\mathrm{m}\,\mathrm{a}^{-1}\,\mathrm{Pa}^{-1}$) result in a too stable GIS (Fig. 7). Only high PDD factors simulate stability similar to values presently observed.





### 3.3 Spin-up: Glacial Greenland ice sheet

A large glacial GIS is created in order to provide reasonable starting conditions for the LIG simulations. This preLIG ice sheet is in equilibrium with the climate forcing (PI NorESM-L precipitation and temperature minus 10 °C, see Sect. 2.3.2) and covers the entire Greenland with ice. The total volume is about 9.1 m SLE for the default basal sliding parameter ($11.2\,\mathrm{m\,a^{-1}\,Pa^{-1}}$)

and atmospheric lapse rate ($6.5\,\mathrm{^\circ C\,km^{-1}}$) (Fig. 8b and simulated ice volume at 135 ka in a). Applying a smaller lapse rate ($5.0\,\mathrm{^\circ C\,km^{-1}}$) gives less of a reduction in the surface temperatures and results in a slightly smaller ice sheet ($\sim 0.2$ m SLE smaller). The opposite occurs when assigning a larger lapse rate ($8.0\,\mathrm{^\circ C\,km^{-1}}$): the resulting ice sheet volume is $\sim 0.2$ m SLE larger. In all cases the climate forcing is mostly below freezing. Varying the PDD factors, which define the amount of surface melt based on temperatures above freezing, has therefore no effect on preLIG ice volume. The ice volume increase from its

modern equivalent does vary, as the modern values do vary with all parameter settings (see Fig. 5).

### 3.4 Last interglacial Greenland ice sheet

The transient LIG GIS simulations are initialized from the equilibrium preLIG ice sheet, and apply the default basal sliding parameter ($11.2\,\mathrm{m\,a^{-1}\,Pa^{-1}}$). Only varying the PDD factors for snow and ice, while keeping the atmospheric lapse rate set to its default value ($6.5\,\mathrm{^\circ C\,km^{-1}}$), results in a large spread of LIG ice sheet evolution (Fig. 8). Melting of the large glacial

ice sheet occurs until a minimum ice sheet is reached at around 126–123 ka (indicated by the dots in Fig. 8). Hereafter the climate forcing gives a positive SMB and the ice sheet expands again. Depending on the PDD factors, the maximum ice loss is 1–6 m SLE with respect to the glacial initial ice sheet. The modern ice sheet volume for the model ensemble is also shown in Fig. 8 (horizontal lines).

For simulations with moderate melting, the minimum LIG GIS volume is larger than its modern GIS equivalent, by up to 0.1–

0.2 m SLE. This only occurs when the PDD factor for ice is at its minimum value ($7\,\mathrm{mm\,water\,day^{-1}\,^\circ C^{-1}}$). Higher ice and snow PDD factors result in a reduction of GIS volume compared to PI. The surface elevation difference between minimum LIG ice sheet configurations and their modern equivalents for the PDD factor simulations are shown in Fig. 9. The LIG minimum ice sheet extents are also indicated (blue contours). Simulated ice is most vulnerable on the southwestern side of Greenland, often resulting in a two dome structure with one large ice sheet covering the north of Greenland and a smaller one in the south, possibly connected on the east coast. The strong melting in the west is found in many ice sheet model reconstructions of the

LIG ice sheet (e.g. Otto-Bliesner et al., 2006; Robinson et al., 2011; Helsen et al., 2013; Calov et al., 2015). In contrast to these and other previous studies, we do not find any melting at the northern rim of the ice sheet. We even simulate an increase in surface height in northern Greenland compared to the PI GIS. Note that this is partly the result of the large glacial ice sheet that initializes the transient LIG simulations. Compared to this initial preLIG GIS, all simulations show reduced surface elevations

at the LIG minimum configuration (not shown).

Another reason for the stability of the ice margin along the northern and eastern coast of Greenland is the relatively cold climate simulated by NorESM-L. Even in the PI simulation the coastal temperatures are below zero, except for southern Greenland (Fig. 2), resulting in positive SMB on the north and east coast. The early LIG climate forcing, albeit several degrees





warmer than PI (Fig. 4), cannot overcome this cold bias, and ice survives along the northern and parts of the eastern coast. This is in contrast to previous studies simulating the LIG GIS. In the most extreme case, Born and Nisancioglu (2012) find a strong ice sheet retreat in the northeast corner of Greenland. In their simulations increased melting in this region cannot be compensated by snow accumulation. The different spatial pattern of LIG GIS reduction between Born and Nisancioglu (2012) and this study is mostly likely due to the different climate forcing, and the use of different boundary and initial conditions.

We furthermore compare the simulated LIG surface elevation for the main ice core locations to available reconstructions from ice cores (Fig. 10). The only Greenland ice core record with continuous information over the LIG is the NEEM record, shown in grey in Fig. 10 (NEEM community members, 2013). The surface elevation changes for the NEEM site are derived from the air content in the ice core, corrected for upstream flow and summer insolation changes (NEEM community members, 2013). As this method includes many assumptions, the uncertainty range is rather large (grey shading in Fig. 10). Still, it does indicate that the surface elevation was higher than today at the start of the LIG, and decreased by 200–300 m during the mid and late LIG to a minimum of $130 \pm 330$ m below the present-day level. Our simulations indicate a similar high elevation at the early LIG ($\sim$130–127 ka), but show a much smaller reduction in surface elevation of only approximately 100 m. This may again suggest a too stable simulated northern GIS. At the other main ice core sites LIG ice is found, but the ice is too disturbed to produce a continuous record covering the LIG (e.g. NGRIP-members, 2004). The oxygen-isotopic composition in these LIG fragments indicate only a minor decrease in surface elevation at Camp Century, NGRIP and GRIP (Johnsen and Vinther, 2007), possibly up to 100 m (NGRIP-members, 2004). At Dye3 a surface elevation lowering of approximately 500 m is proposed (NGRIP-members, 2004; Johnsen and Vintner, 2007). Unfortunately these minima might not have occurred simultaneously, and timing is difficult to reconstruct for the disturbed ice cores. We therefore indicate the maximum likely elevation changes as grey shading in Fig. 10. The simulated surface elevation difference at EGRIP is included for sake of completion, but as it is not yet drilled, currently no data is available for comparison at this location. Comparing the ensemble of LIG simulations to the other ice core elevation data, it is clear that five simulations largely overestimate the surface lowering at GRIP, with elevation changes of more than 500 m. The same simulations also indicate extensive melt at NGRIP and Dye3, and in some cases melt the entire ice at these three most southerly cores. With LIG ice sheet volumes of 5.1 m SLE or less, a large part of ice in the southwest disappears (see Fig. 9, simulations indicated by red shading). The large negative SMB in these simulations is caused by the relatively high PDD factors for snow and ice (5 and 8 mm water day$^{-1}\,^\circ$C$^{-1}$ for snow, and 15 and 20 mm water day$^{-1}\,^\circ$C$^{-1}$ for ice). Omitting these unrealistic simulations reduces the maximum LIG Greenland ice loss to 3.2 m SLE or 1.8 m SLE compared to the preLIG and PI ice volumes, respectively (solid lines in Fig. 8).

The suite of experiments simulating a modern GIS show that for low PDD factors the ice sheet is too stable, with more than 65% of the accumulation used for the SMB (blue shading in Fig. 9, see also Sect. 3.2). This restricts the minimum amount of LIG GIS melt. Only one simulation fits both the stability criteria and the maximum elevation loss estimated at the ice core sites (accentuated line in Fig. 8). This simulation has PDD factors of 8 mm water day$^{-1}\,^\circ$C$^{-1}$ for snow and 10 mm water day$^{-1}\,^\circ$C$^{-1}$ for ice. Its maximum LIG ice loss is about 2.2 m SLE compared to the initial glacial ice sheet and approximately 0.8 m SLE compared to its modern ice sheet volume (Fig. 7, 8 and 9).





In the previous comparison only the PDD factors were varied, keeping the temperature lapse rate to its default value ($6.5\,^\circ\mathrm{C\,km^{-1}}$). However the temperature lapse rate does also affect the ice volume evolution during the LIG. Figure 11 shows this for the preferred simulation of the PDD factor analysis (PDD factors of 8 and $10\,\mathrm{mm\,water\,day^{-1}\,^\circ C^{-1}}$ for snow and ice, respectively). For temperature lapse rates smaller than default, the SMB/acc ratio is too high, and these simulations are omit-

ted. The stability criteria is fulfilled for simulations with temperature lapse rates between 6.5 and $8.0\,^\circ\mathrm{C\,km^{-1}}$ (see Fig. 11c for surface elevation maps). The resulting maximum LIG ice loss estimate is between 0.8–2.2 m compared to the simulated modern ice sheet volume. This range is similar to other recent model studies (e.g. 1.2–3.5 m of Helsen et al. (2013); 0.4–3.8 m of Stone et al. (2013); 0.6–2.5 m of Calov et al. (2015)). The timing of the maximum contribution of GIS melt to the LIG sea-level highstand is between 124 and 122 ka.

## 4 Discussion

### 4.1 NorESM-L climate forcing

#### 4.1.1 Comparison to other simulated LIG climates

The climate forcing largely defines the shape and extent of the simulated GIS. On average NorESM-L computes lower temperatures than observed along the coast of Greenland, particularly in the north (Fig. 2). However, in the south and in the central

regions, simulated temperatures are warmer than observed. The main reason for this warm bias is the relatively low surface topography over central and southern Greenland defined in NorESM-L.

We compare our results to the annual and summer (June-July-August) mean PI temperatures of the 13 models used in the model intercomparison study of Lunt et al. (2013). This compilation includes our NorESM-L simulations. Most models simulate a general temperature pattern over Greenland somewhat similar to observed, with low temperatures in central Greenland

and higher temperatures on the southern coast (model mean shown in Fig. 12a&b). However, the model simulations show regional cold and warm biases of up to $10\,^\circ\mathrm{C}$ compared to observations. For the reduced complexity models, LOVECLIM and CLIMBER, the simulated modern day warm bias over southeast and central Greenland is even larger. The largest cold bias is simulated by the Kiel Climate Model (KCM), which simulates PI temperatures more than $10\,^\circ\mathrm{C}$ lower than observed on the east coast. The spatial pattern of the standard deviation of the simulated temperatures is dominated by the biases of these three

model simulations, and shows maximum values of up to $6\,^\circ\mathrm{C}$ over central and southeast Greenland (Fig. 12c&d). Interestingly, simulated annual mean temperatures are less consistent between the models than the summer mean (compare Fig. 12c to d). When compared to the mean of the investigated models, NorESM-L shows a cold bias over northern Greenland and a warm bias over the southern tip of Greenland (Fig. 12e&f), as expected from the comparison to the observational data (Fig. 2).

From the model intercomparison of Lunt et al. (2013) 8 models simulations are available for 125 ka, and 7 models for

130 ka. Figure 13 shows the model mean annual and summer temperature increase for 125 and 130 ka compared to the PI. Again, these averages include the NorESM-L simulations. A similar pattern is found for the 125 and 130 ka model mean temperature increase. The annual mean temperature increase at 125 ka for NorESM-L is smaller than in the other models





(compare Fig. 4 and Fig.13). However, similar to NorESM-L, most models compute a larger temperature increase over northern than over southern Greenland. For 130 ka, NorESM-L simulates a small annual mean cooling compared to PI (Fig. 4), which is in contrast to most of the other models. The other cold exception, CSIRO, simulates a similar cooling over large parts of Greenland, with a small warming of approximately 1 °C in the centre. In contrast, CCSM3 simulations performed by NCAR

show a much larger annual mean warming than the model mean (Greenland mean warming of 2.6 and 3.6 °C for 125 and 130 ka, respectively, compared to model mean of 1.1 °C).

For 125 and 130 ka the model mean summer temperature increase is approximately 2.8 and 2.5 °C, respectively. Largest increase is found in central (east) Greenland. This pattern and magnitude is also reproduced by NorESM-L (Fig. 4). Even though the model mean temperature patterns are similar to NorESM-L, the individual model temperature increases are quite

different. For example, again CCSM3 simulations performed by NCAR compute a more extreme summer warming during the early LIG (Greenland mean summer warming of 4.0 and 4.5 °C for 125 and 130 ka, respectively, compared to model means of 2.8 and 2.5 °C). In contrast CLIMBER simulates a relatively small summer temperature increase with only 1.7 and 2.0 °C warming for 125 and 130 ka compared to PI.

The differences between the simulated climates are partly due to different model dynamics and complexity, but also due to

small variations in model forcing (e.g. greenhouse gases) and set-up (e.g. dynamic vegetation or not). Climate forcing taken from models with higher 125 and 130 ka temperatures (especially summer temperatures) will melt the GIS more. It is difficult to estimate the exact effect, because with a different climate forcing the procedure of testing the uncertain model parameters should be repeated. A thorough assessment of the LIG climate over Greenland and reduction of the possible range could largely improve our confidence in the resulting minimum ice sheet configurations, but is beyond the scope of this study.

### 4.1.2 Direct vs anomaly forcing

Biases in the simulated PI climate are often circumvented by considering the simulated *change* in climate, and adding this as an anomaly on top of observations ('anomaly forcing'). The rationale behind is that the offset is model dependent and time independent, i.e. the same offset occurs in both time slices and therefore cancels out. In this study we used the simulated climates directly ('direct forcing'). A disadvantage is that NorESM-L's cold bias in the northern North Atlantic is therefore

present in all simulations, causing too much ice over northern Greenland. A quick test using the anomaly method results in less ice over northern Greenland (not shown). The advantages of using direct forcing is that dynamically consistent changes to the atmospheric circulation in the different climate states are considered, no biases are hidden, and the results can be discussed by comparing to the forcing directly.

### 4.1.3 Effect of modern GIS in NorESM-L

The GIS used as a boundary condition for the simulations with NorESM-L is fixed to represent the present-day ice sheet. However, during the LIG the GIS is reduced by approximately 1–2 m SLE. A smaller, lower ice sheet topography would in turn affect the climate. As our climate and ice sheet models are not coupled and NorESM-L does not consider a dynamic ice sheet, we cannot account for this feedback. However, a smaller ice sheet should result in a warmer climate, therefore further





enhancing ice sheet melt. Stone et al. (2013) compared LIG summer temperatures over Greenland from GCM simulations with and without a GIS. The absence of an ice sheet increases the Greenland temperatures by over $10\,^{\circ}\mathrm{C}$ (Stone et al., 2013). This is mostly due to the lowering of the surface topography. They show that temperatures from the GCM with a modern GIS configuration are much closer to the reconstructed temperatures, and conclude this to be additional evidence of the GIS largely

surviving LIG warmth. It is therefore likely that updating NorESM-L with a slightly reduced GIS would probably have only a minor effect on the estimated minimum ice sheet configuration. In order to properly assess the effects of the changing GIS on the climate, the two models should be directly coupled. However, unfortunately, a fully coupled model set-up is at present too computationally expensive.

## 4.2   Atmospheric temperature lapse rate

The observed global mean temperature lapse rate is approximately $6.5\,^{\circ}\mathrm{C\,km^{-1}}$. How much temperature changes with elevation strongly depends on the moisture content of the air. In regions where air is saturated with water vapour, lapse rates are typically lower than this mean. In contrast, in regions where air is dry or unsaturated, the lapse rate can be as high as $9.8\,^{\circ}\mathrm{C\,km^{-1}}$. This means that the lapse rate varies from location to location, and is also different depending on the season. Fausto et al. (2009) compute an annual mean lapse rate of $6.8\,^{\circ}\mathrm{C\,km^{-1}}$ using Automatic Weather Station (AWS) data from

seven transects over the GIS. They observed winter and summer lapse rates of about $2$–$3\,^{\circ}\mathrm{C\,km^{-1}}$ higher and lower, respectively. Unfortunately these transects did not cover the, in general dry, northeastern Greenland. A new study with a much higher data density, including data from the northeast and central Greenland, shows temperature lapse rates between 7 and $8\,^{\circ}\mathrm{C\,km^{-1}}$ (A.-K. Faber, personal communication, 2015). For the lapse rate computation from this newest dataset, borehole temperatures from the interior of the ice sheet were included. This increases the number of temperature observations in the computation. On

the other hand, this might cause an overestimation of the resulting lapse rate as we compare temperatures in the firn layer at high elevations to $2\,\mathrm{m}$ temperatures at lower elevations. The firn temperature can be up to $2\,^{\circ}\mathrm{C}$ different from the annual mean air temperature (Steffen and Box, 2001). Several studies show lower temperatures at the eastern side of Greenland compared to the western side given the same elevation, indicating a higher temperature lapse rate in the east compared to the west (e.g. Steffen and Box, 2001; Fausto et al., 2009). Our accepted range of temperature lapse rates (6.5 to $8.0\,^{\circ}\mathrm{C\,km^{-1}}$) is similar to

the annual mean values over Greenland from these studies. Unfortunately, we do not know how the lapse rate changed over time. To include seasonal and spatial variations for the lapse rates applied in this study is therefore deemed inappropriate until additional data becomes available. The lapse rate chosen does have a large effect on the resulting simulated ice volume loss. In this study we show that increasing the lapse rate from 6.5 to $8.0\,^{\circ}\mathrm{C\,km^{-1}}$ results in an additional ice loss of $1.4\,\mathrm{m}\,\mathrm{SLE}$ for Greenland (2.2 compared to $0.8\,\mathrm{m}\,\mathrm{SLE}$).

## 4.3   Glacial ice sheet size

Our preLIG ice sheet has a volume of approximately $9.1\,\mathrm{m}\,\mathrm{SLE}$. Unfortunately it is not possible to evaluate the modelled preLIG ice volume with direct paleoclimatic reconstructions, as the existing Greenland ice core records do not cover the glacial period preceeding the LIG. Helsen et al. (2013) simulate their time-dependent preLIG glacial Greenland ice volume by





reconstructing a Greenland climate forcing based on the Vostok record from Antarctica. While taking a very different approach, their derived ice sheet volume at the start of the LIG of approximately 9 m SLE is very similar to our preLIG estimate. Stone et al. (2013) also initialise their LIG simulations with a glacial GIS. They obtain a preLIG ice sheet by applying a constant glacial climate from the general circulation model HadCM3 to an ice sheet model in its modern state. A spin-up procedure
makes sure that the ice sheet model is in equilibrium with the cold 136 ka climate forcing. The preLIG ice sheet volume of Stone et al. (2013) varies between ~8.6 and ~10.5 m SLE for their accepted simulations. Our 9.1 m SLE also falls within this range.

Model studies focusing on the last glacial maximum (LGM) also simulate a glacial GIS of approximately 9 m SLE when considering the modern Greenland land area (e.g. Fyke et al., 2011). However, geomorphological reconstructions indicate that
the GIS extended well onto the continental shelf (e.g. Funder et al., 2011), and recent estimates of the LGM Greenland ice volume are closer to 11–12 m SLE (Simpson et al., 2009; Lecavalier et al., 2014). It is possible that the preLIG GIS was as large as or larger than the LGM GIS. However, data is sparse and inconclusive (e.g. Funder et al., 2011; Vasskog et al., 2015). In the current set-up of the ice sheet model, the land-sea mask is based on the present-day land area, and continental shelves are omitted. Future research assessing the preLIG GIS should include an extended, updated Greenland land area.

If LIG simulations are initiated from an even larger GIS, it will take more time to melt away this extra ice. The climate forcing might be adjusted in the future, but the turnover from a warmer to a colder climate is relatively well defined, as this is time-locked by the well-known variations in orbital configuration. Starting from a larger preLIG ice sheet, while having the same amount of time for melting, would therefore result in a smaller reduction of LIG ice (compared to its PI state).

### 4.4 Depositional location of LIG ice

LIG ice assessed in the deep ice cores did not originate from snow falling on the present-day location of the ice core, but from snow falling at the LIG location of the ice core. This location could have been a few hundred kilometres upstream from the present location for ice cores located away from the ice divide (e.g. NGRIP, NEEM, Camp Century, Dye3). The summit ice cores (e.g. GISP, GRIP) probably did not shift location much over time, as the horizontal ice velocities are very small in the interior of the ice sheet. In contrast, the 128 ka old ice at NEEM is though to be deposited approximately 200 km further
up the dome (NEEM community members, 2013). Similarly, Camp Century and Dye3 shifted location over time. Climate information imprinted in the ice therefore originates from a different location than the present location of the ice cores. Hence, the paleo constraint of maximum surface elevation change compares ice elevations from the LIG upstream location to the present-day elevation at the ice core. The upstream location of Dye3 during the LIG is likely approximately 100 km further to the (south)west, at the south dome. Ice surface elevations are higher there than at Dye3. During the LIG most of the southern
dome had a lower elevation (Fig. 9), but still the top of the dome was higher than the LIG elevation at the present-day location of Dye3. The computation of maximum surface elevation anomaly as simulated for Dye3 (Fig. 10) does not correct for the shift in deposition location and therefore overestimates the amount of surface lowering. In other words, some simulations that were rejected based on a too large surface elevation anomaly between LIG and today, might be acceptable if you compare the upstream paleo elevation to the present-day elevation. This would suggest a larger possible melt of the LIG GIS, and a larger





contribution to the LIG highstand. The exact upstream location of the ice cores can be traced by using the horizontal velocities of ice between the LIG and PI. Unfortunately we did not simulate the entire glacial cycle, and therefore do not have velocity information between the LIG and PI. Future studies should consider such an assessment of (paleo) elevations, especially if the uncertainties on the elevation changes derived from the ice core records become smaller.

## 4.5 Other uncertainties and future work

The LIG results presented in this study largely depend on the climate forcing and initial (preLIG) conditions (discussed above). However, also other parts of the model set-up and initialization will affect the results, such as the initial bedrock topography, geothermal heat flux, and the PDD method. The latter underestimates the surface amount of melt, because the direct effect of summer insolation (the part not considered though the ambient temperatures) and related non-linear feedbacks are not included (van de Berg et al., 2011). In order to account for this missing effect, future GIS simulations could include a more sophisticated SMB model, such as the insolation-temperature-melt model used by Robinson et al. (2011). When such a SMB model is included, a higher summer insolation over northern compared to southern Greenland could cause additional melting in the north, possibly changing the shape of the remaining GIS and raising the likely contribution of GIS melt to LIG sea-level rise. The amount of melting in southwestern Greenland (main region of melt in this study) is, however, strongly restricted by the maximum LIG surface elevation lowering compared to PI estimated from ice core data.

In order to better understand the various estimates of LIG GIS volume, and to further decrease the range of these estimates, several thorough model and model-data intercomparisons are needed. A higher number of temperature proxy records at and around Greenland are needed to better evaluate the range of simulated LIG climates. The resulting 'best' climate can then be used as ice sheet model forcing for a better assessment of the current spread of GIS simulations.

## 5 Conclusions

We used simulated temperature and precipitation patterns from the NorESM-L climate model for the pre-industrial (PI) as well as four last interglacial (LIG) equilibrium timeslices to assess the volume, extent and stability of the Greenland ice sheet (GIS). Our model ensemble includes a range of basal sliding parameters, atmospheric temperature lapse rates and melt factors for snow and ice. The LIG simulations were initialized by a large, glacial ice sheet of approximately 9 m SLE. We used 1) the present-day GIS volume and ice elevations (Bamber et al., 2013), 2) the surface mass balance to accumulation ratio (a stability criteria), 3) the presence of LIG ice at all deep ice core locations, and 4) a maximum surface elevation reduction of 100 m between LIG and today at the deep ice core locations (500 m for Dye3) to constrain the simulated GIS size.

We find a maximum GIS reduction of 0.8–2.2 m SLE compared to the simulated PI/modern ice sheet volume. Most of the melt occurs in the southwestern part of Greenland. This melt causes a two-dome structure with a large ice dome over northern Greenland and a small ice dome covering the south. The timing of the maximum ice melt over Greenland is estimated to have occurred between 124 and 122 ka.



We suggest a default basal sliding parameter ($11.2\,\mathrm{m\,a^{-1}\,Pa^{-1}}$), relatively high melt factors (8 and $10\,\mathrm{mm\,water\,day^{-1}\,{}^\circ C^{-1}}$ for snow and ice, respectively), and an average to high atmospheric temperature lapse rate ($6.5\text{--}8\,\mathrm{{}^\circ C\,km^{-1}}$) as preferred SICOPOLIS ice sheet model set-up.

*Acknowledgements.* We thank Ralf Greve for making the SICOPOLIS ice sheet model publicly availability and well documented. We also thank Andreas Born for his advice and comments during the initial phase of this study. This project benefitted from support from Past4Future 'Climate change - Learning from the past climate', a Collaborative Project under the 7th Framework Programme of the European Commission (grant agreement no. 243908). P.M.L. is funded by the Research Council of Norway through the IceBed project (221598). K.H.N. is funded by the European Research Council under the European Community's Seventh Framework Programme (FP7/2007-2013) / ERC grant agreement 610055 as part of the Ice2Ice project.





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





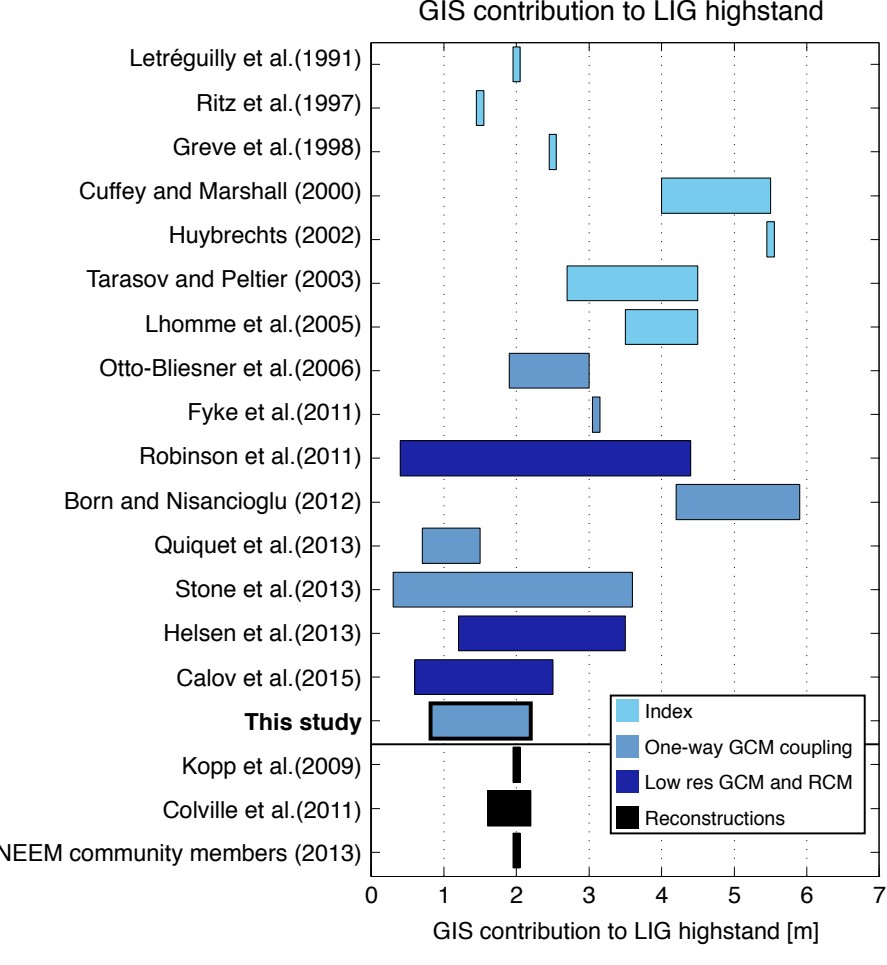

**Figure 1.** GIS contribution to the LIG highstand compared to present-day from ice sheet model simulations (blue) and reconstructions (black). The different shades of blue indicate the type of climate forcing: index method (light blue), one-way GCM coupling (blue) or a combination of a low resolution GCM and RCM (dark blue).





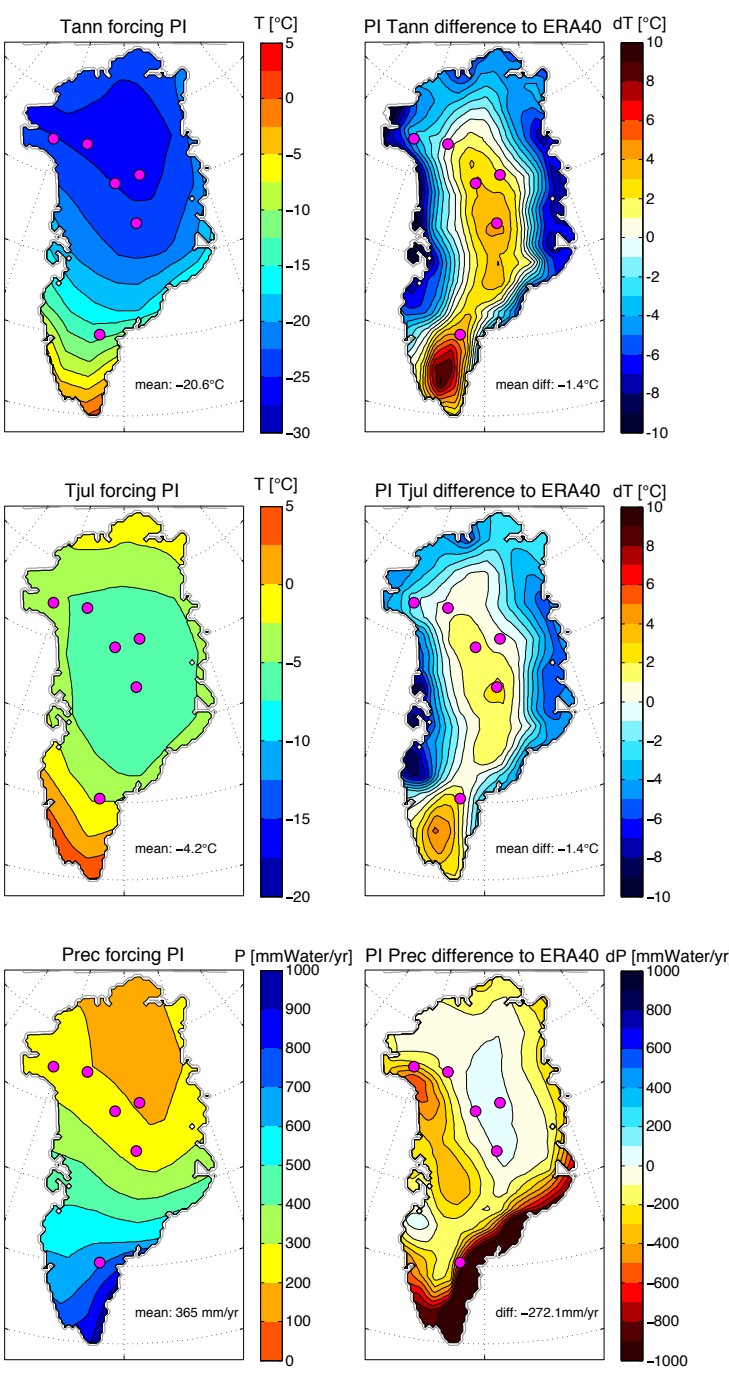

**Figure 2.** PI climate forcing over Greenland and its difference to observations. Tann (upper row), Tjul (middle row), and precipitation (lower row) as simulated by NorESM-L (left) and compared to ERA40 data (right). Mean values and mean differences over Greenland are stated in the plots. Purple dots indicate deep ice core locations, from north to south: Camp Century, NEEM, EGRIP, NGRIP, GRIP and Dye3.





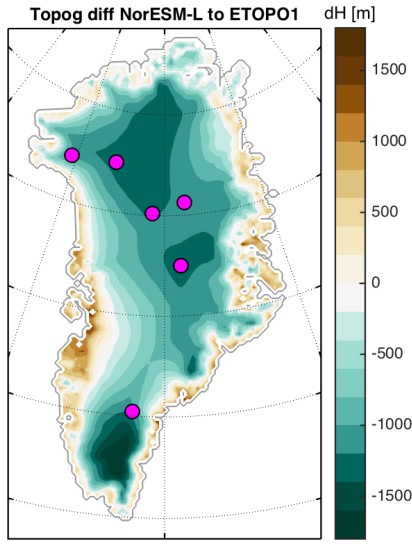

**Figure 3.** Difference in surface elevations over Greenland between topography set in NorESM-L and topography observed (from ETOPO1 dataset).

**Table 1.** Scheme of orbital and greenhouse gas forcing applied in the PI and LIG simulations.

| Exp. name/ | Orbital parameters | | | Greenhouse gas concentrations | | |
|---|---|---|---|---|---|---|
| Time slice | Ecc | Obl [°] | Peri-180 [°] | $CO_2$ [ppm] | $CH_4$ [ppb] | $N_2O$ [ppb] |
| PI | 0.0167 | 23.45 | 102.0 | 280 | 760 | 270 |
| 115 ka | 0.0414 | 22.41 | 110.9 | 280 | 760 | 270 |
| 120 ka | 0.0411 | 23.01 | 28.0 | 280 | 760 | 270 |
| 125 ka | 0.0400 | 23.80 | 307.1 | 276 | 640 | 263 |
| 130 ka | 0.0382 | 24.24 | 228.3 | 257 | 512 | 239 |

**Table 2.** Default values and range of uncertain parameters assessed in the ice sheet simulations.

| Parameter | Default value | Other values tested | Unit |
|---|---|---|---|
| Basal sliding | 11.2 | 5, 17 | $m\,a^{-1}\,Pa^{-1}$ |
| Atmospheric temperature lapse rate | 6.5 | 5, 8 | $°C\,km^{-1}$ |
| PDD factor for snow | 3 | 5, 8 | $mm\,water\,day^{-1}\,°C^{-1}$ |
| PDD factor for ice | 7–15 | 7, 10, 15, 20 | $mm\,water\,day^{-1}\,°C^{-1}$ |





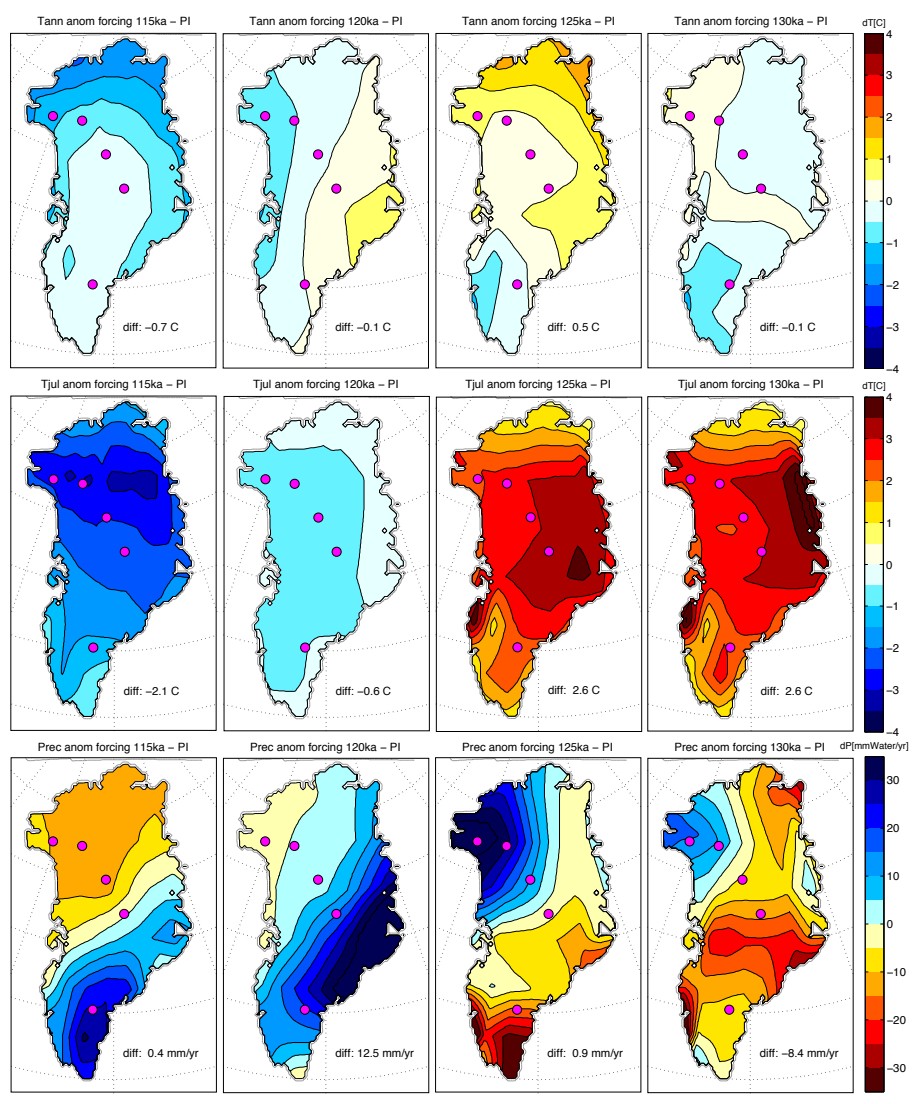

**Figure 4.** LIG climate anomaly compared to PI for Tann (upper row), Tjul (middle row) and precipitation (lower row). From left to right: 115 ka–PI, 120 ka–PI, 125 ka–PI, and 130 ka–PI. Mean difference to PI over Greenland is shown in figures. Purple dots indicate deep ice core locations.





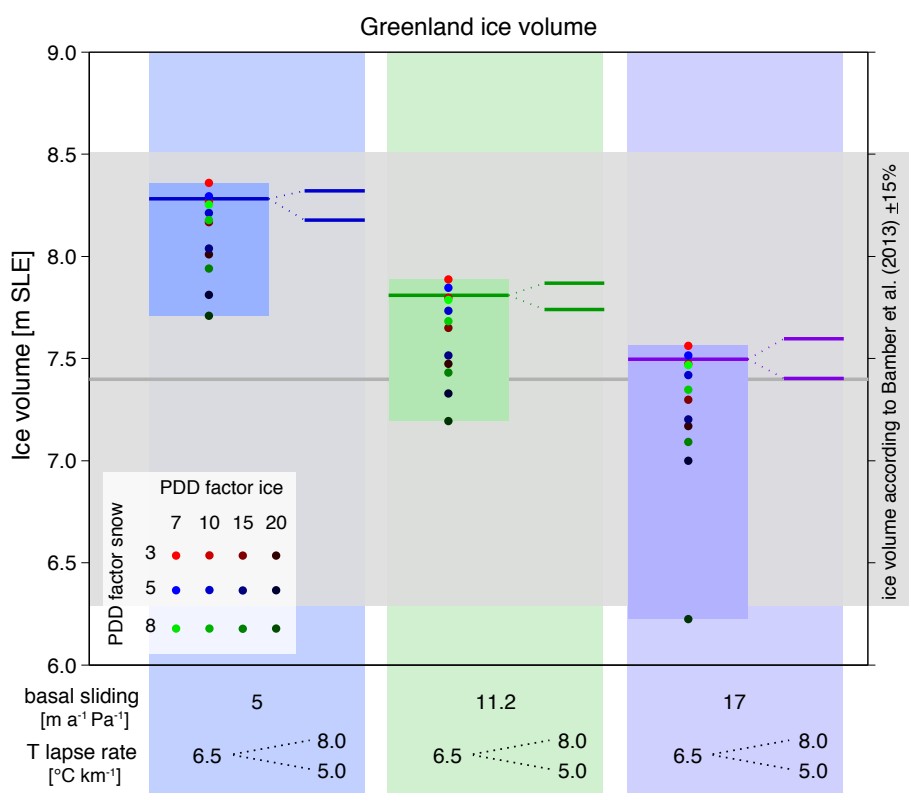

**Figure 5.** PI Greenland ice volume for the ensemble simulations. Coloured columns indicate different basal sliding parameters: 5 (blue), 11.2 (default, green), and $17\,\mathrm{m\,a^{-1}\,Pa^{-1}}$ (purple). Dots represent PDD factors of snow and ice. Horizontal lines indicate changes in atmospheric lapse rate: 6.5 (default, centre line), 5.0 (lower line) and $8.0\,^{\circ}\mathrm{C\,km^{-1}}$ (upper line). Grey horizontal line indicates the observed present-day GIS volume according to Bamber et al. (2013), the grey shading a 15% offset.





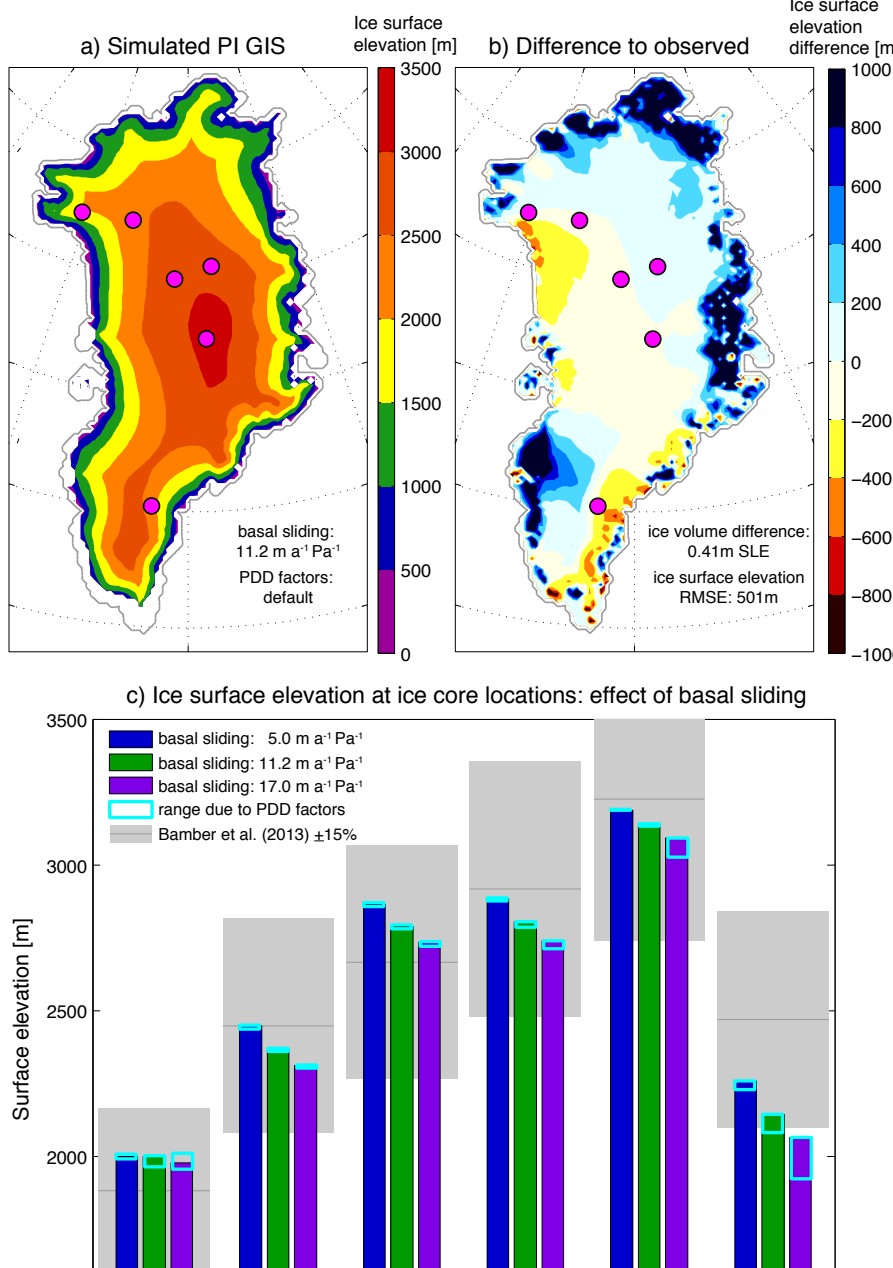

**Figure 6.** Simulated PI surface elevations compared to observations. a) Map of simulated PI surface elevation using all default parameters. b) Difference of a) compared to the surface elevations from observations (Bamber et al., 2013). Ice volume difference and root mean square error (RMSE) in surface elevation are given. Purple dots indicate deep ice core locations. c) Simulated and observed surface elevation at deep ice core locations. Columns show basal sliding parameter: 5 (blue), 11.2 (default, green), and $17\,\mathrm{m\,a^{-1}\,Pa^{-1}}$ (purple). Light blue boxes indicate range due to the range of PDD factors. Grey horizontal line and shading show observed surface elevation and 15% offset.





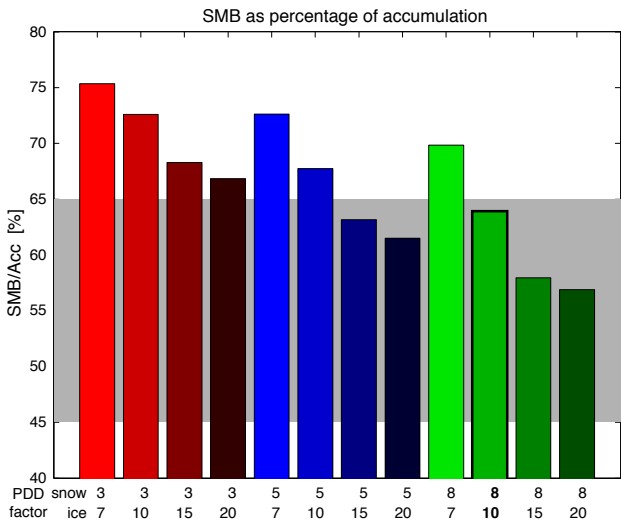

**Figure 7.** Stability of the PI GIS for a range of PDD factors and default basal sliding and atmospheric temperature lapse rate. Grey shading illustrates the accepted range of SMB/accumulation. Values above this range point to a too stable GIS, whereas values below indicate a too unstable GIS. Our preferred simulation (PDD factors of 8 and $10\,\mathrm{mm\,water\,day^{-1}\,^{\circ}C^{-1}}$ for snow and ice, respectively) is emphasised by a black rim.





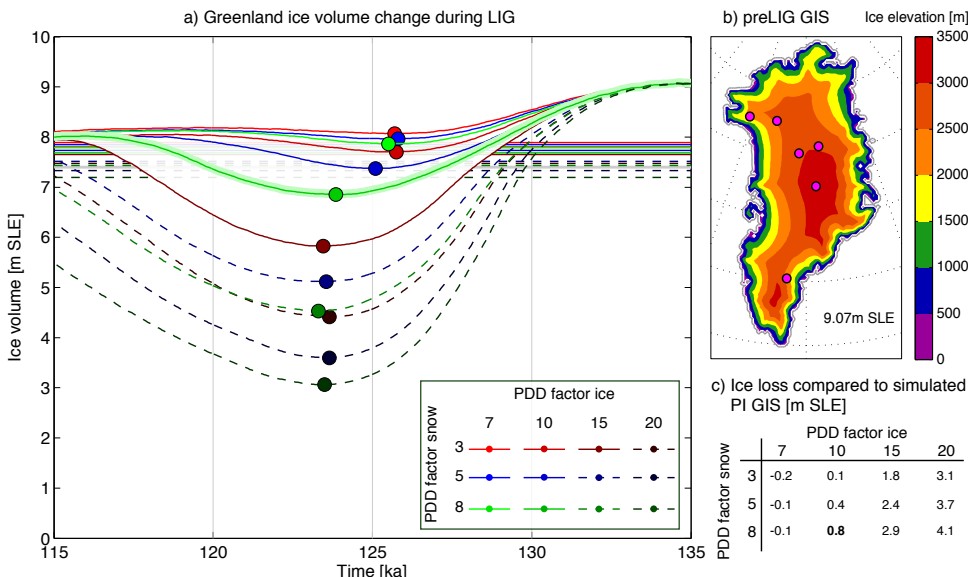

**Figure 8.** Greenland ice volume loss over the LIG. a) Evolution of LIG Greenland ice volume for a range of PDD factors. Basal sliding and atmospheric temperature lapse rate are at default values. Horizontal lines in background show the PI equivalent ice volumes. Dots indicate the time and volume of the minimum GIS configuration for each LIG simulation. Dashed lines show simulations with too much melt (see also red shaded plots in Fig. 9). Solid lines indicate simulations with acceptable surface elevation changes (blue and white shading in Fig. 9). The ice volume evolution of our preferred simulation (with PDD factors of 8 and $10\,\mathrm{mm\,water\,day^{-1}\,{}^\circ C^{-1}}$ for snow and ice, respectively) is emphasized with light green shading. b) Map of the ice surface elevations of the preLIG GIS, the initial elevations for the LIG simulations. Mean ice volume is stated in plot. Purple dots indicate deep ice core locations. c) Maximum LIG ice loss compared to simulated PI values for the suite of PDD factors. Our preferred simulation is indicated in bold.





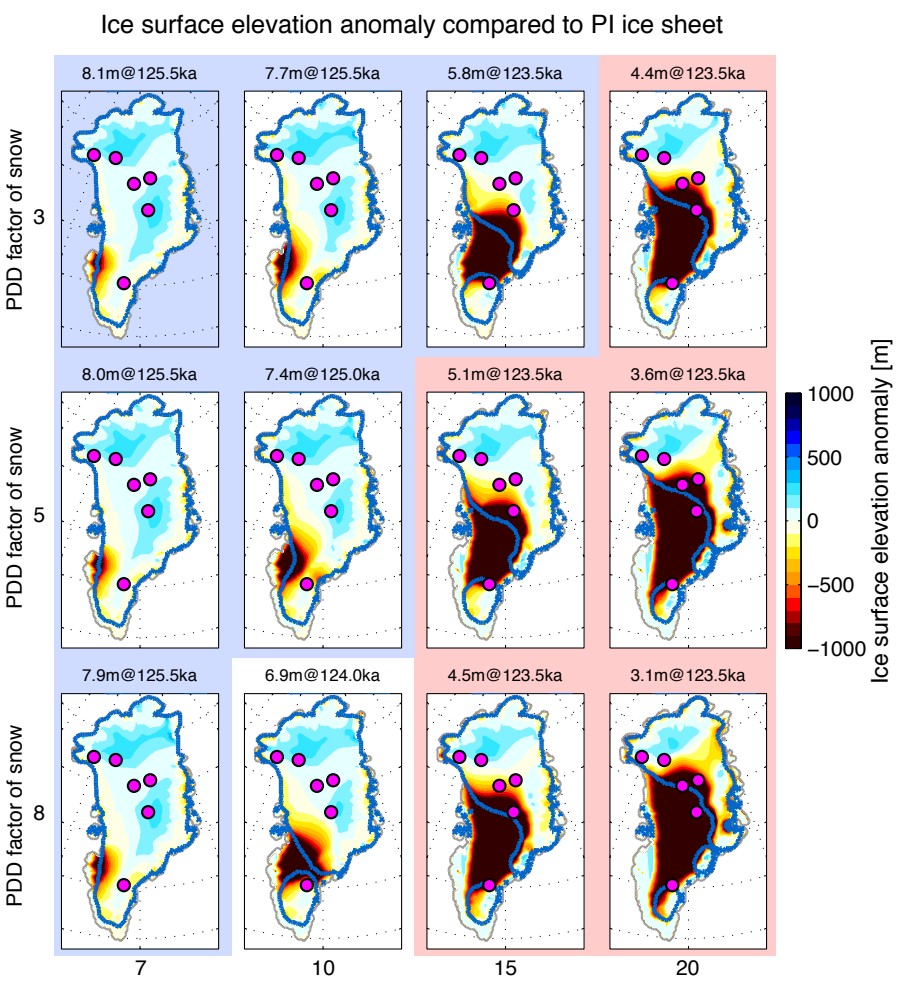

**Figure 9.** LIG ice surface anomalies to PI for the minimum GIS configurations, for a range of PDD factors. Blue contour indicates the LIG ice sheet extent. Minimum GIS volume and the timing of the minimum are stated above each panel. Blue shading emphasizes too stable ice sheet configurations, whereas red shading shows simulations with too much melt (i.e. too large reduction in surface elevation). The simulation without shading is our preferred solution. Purple dots indicate deep ice core locations.





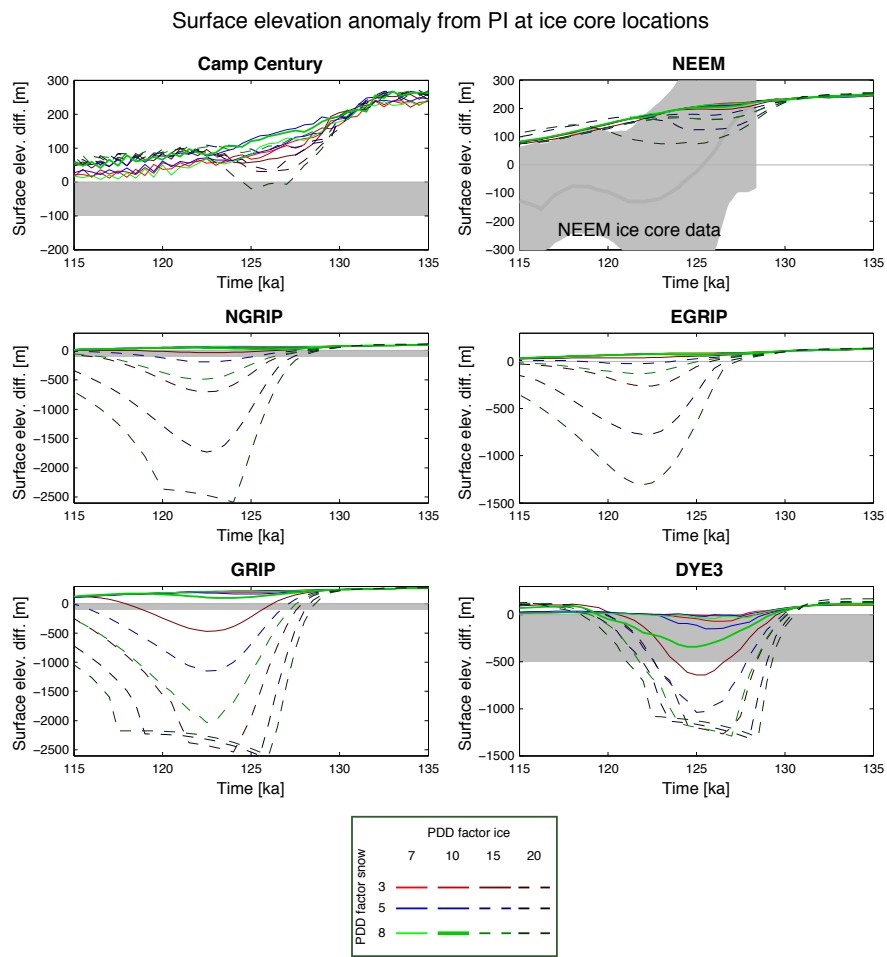

**Figure 10.** LIG surface elevation anomaly to PI at the deep ice core locations compared to reconstructions. Coloured lines show simulations with different PDD factors, where dashed (solid) lines indicate rejected (accepted) simulations with much more (similar or less) surface melt than reconstructed from the ice core records. Grey shading illustrates the likely maximum surface elevation reduction as reconstructed from the ice core records.





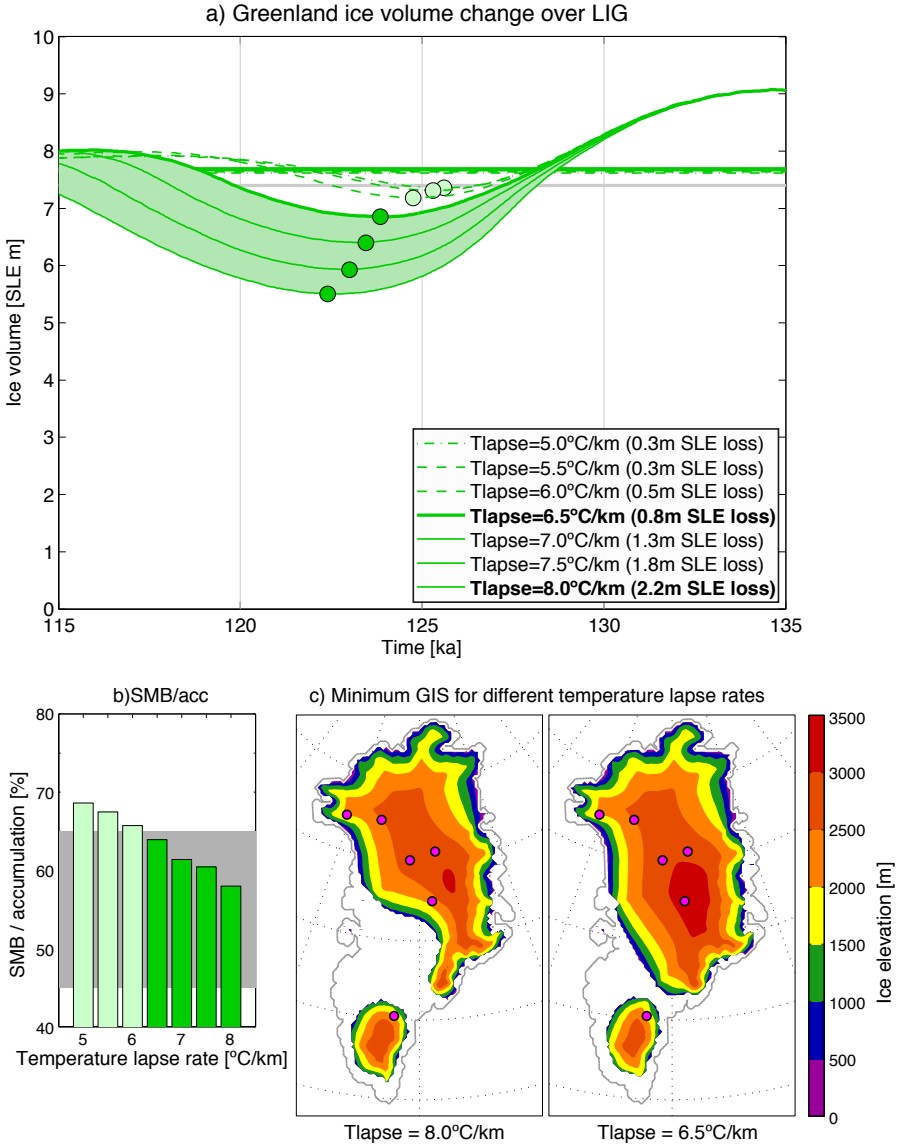

**Figure 11.** a) LIG GIS volume change compared to PI for the best PDD factors (8 and $10\,\mathrm{mm\,water\,day^{-1}\,^\circ C^{-1}}$ for snow and ice, respectively) and a range of atmospheric temperature lapse rates. Solid (dashed) lines represent accepted (rejected) simulations based also on the stability criteria in b). Labels state maximum amount of volume loss for each simulation. Dots indicate when the minimum ice sheet is reached. Horizontal green lines show equivalent PI ice volumes. The observed GIS volume is shown as grey line. b) Stability criteria for the set of simulations of a). Grey shading illustrates accepted range. c) Maps of ice surface elevation for the two extremes of the accepted simulations of a&b). Purple dots indicate deep ice core locations.





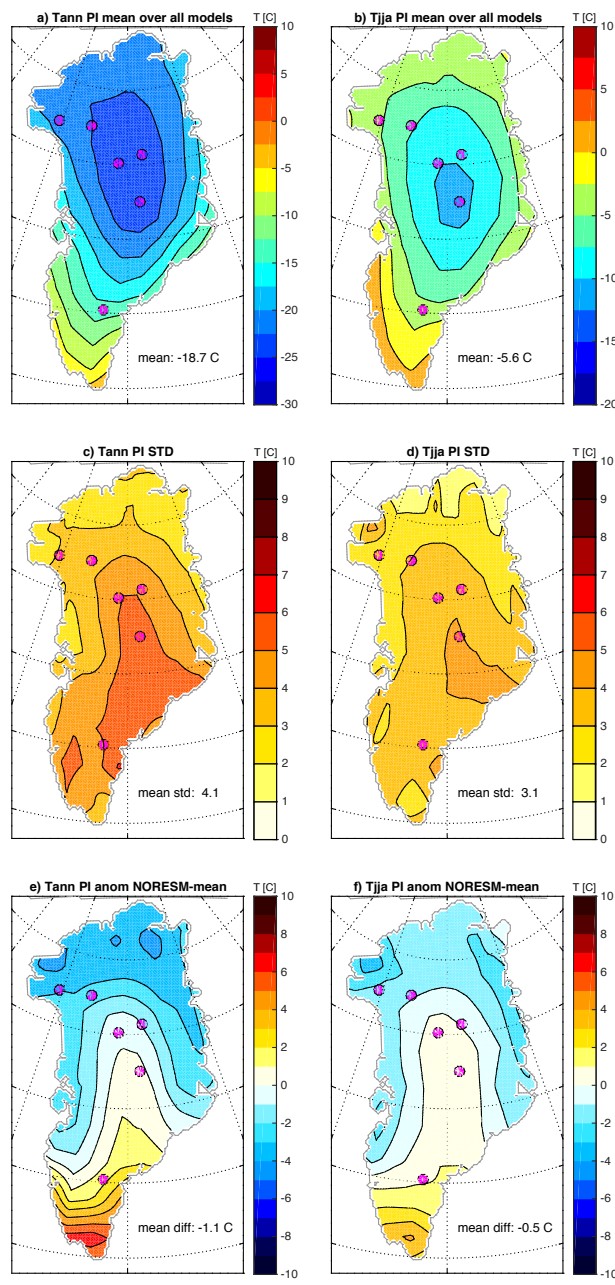

**Figure 12.** Comparison of PI temperatures over Greenland from 13 climate model simulations (Lunt et al., 2013). a) ensemble annual mean temperature (Tann), b) ensemble summer mean temperature (Tjja) , c) Tann standard deviation for ensemble, d) Tjja standard deviation for ensemble, e) Tann anomaly NorESM to ensemble mean, and f) Tjja anomaly NorESM to ensemble mean. Purple dots indicate deep ice core locations.





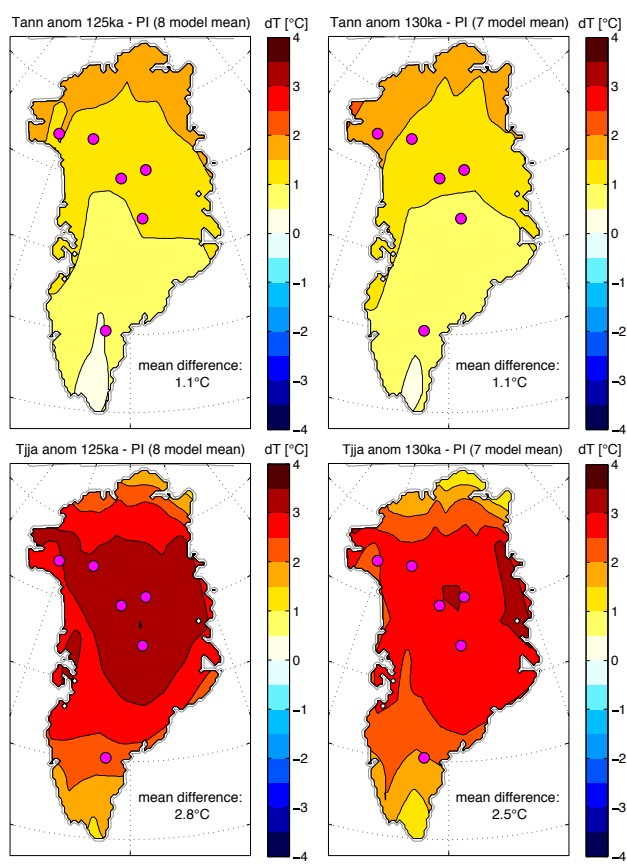

**Figure 13.** Maps of mean LIG Greenland temperature change compared to PI from the climate model intercomparison of Lunt et al. (2013). Upper (lower) row shows Tann (Tjja). Mean over 125 ka–PI is computed from 8 models (left), whereas the mean over 130 ka–PI is calculated from 7 models (right). The 8/7 model and Greenland mean difference between LIG and PI is stated in each plot. Purple dots indicate deep ice core locations.