# Peer review of "Moderate Greenland ice sheet melt during the last interglacial constrained by present-day observations and paleo ice core reconstructions"

_The Cryosphere, 2016_

## Referee Comment (RC1) · Anonymous Referee #1 · 8 Apr 2016

The authors present a study of the evolution of the Greenland Ice Sheet (GIS) during the Last Interglacial (LIG), using time slices from the GCM NorESM and ice model SICOPOLIS and select the most suitable model outcome using extend and local elevation constraints from ice core data.

I firstly make my apologies for the long time I've needed to complete this review, and secondly make my apologies that I'm not very positive concerning the research presented. The authors use the PDD (positive degree day) method, which is not applicable here, and the results present never look that realistic that it earns some reliability. The

latter (an ice sheet glued to the Arctic Ocean) could be bad luck, but using the PDD is a method flaw that could have been avoided. To my opinion the results presented in this manuscript does not provide any new information on the LIG climate, LIG GIS extend or the GIS sensitivity to climate change, nor in how we could tackle this problem.

In the cryospheric community, we are generally kind, but by using a method (PDD) that is unsuitable, it leaves me with no other option that to advise the editor to reject this manuscript unless the authors redo the ice model simulations using an ITM method to derive melt rates from the GCM output.

**Primary concerns**

*Experiment setup: PDD*

As the authors know, the PDD is the traditional "way out" if someone has no surface energy balance (SEB) data to estimate melt and runoff. It does work after careful tuning – yes I know - if cloud and insolation characteristics remain similar. Hence, it works for each glacier individually for time scales up to a few centuries. For multi-millennia simulations over the entire GIS the assumption of two constant PDD factors for snow and ice does not hold, neither spatially nor in time. Bluntly said, as PDD is not valid here, it should not have been used. Period. And as melt is the key process that drives the evolution of the GIS during the LIG, it's lethal for this manuscript.

The (for me obvious) method to use is the an Insolation-Temperature Method (ITM, e.q. Robison et al, 2011, Climate of the Past) which explicitly includes insolation into the derivation of melt/runoff. And also an ITM allows tuning until the model results start to resemble past and current GIS states. Honestly, I can't understand why the authors did not take the effort to implement this. It is not complicated, neither state of the art. A state of the art approach would be a method like presented in Vizcaino and others,

J.Clim., 2013. There is no longer an excuse for using PDD.

*Model performance of PI & LIG climate*

Figure 6a and b does not give a very affirming feeling that the NorESM & SICOPOLIS combination can realistically model the GIS. The figures show that the model let the pre-industrial (PI) ice sheet grow until it meets the shore. (As SICOPOLIS does not model ice shelves, it can't grow into the sea but if it could, it even might have grown on the continental shelf). Figure 6ab says me that melt and runoff are largely underestimated for the default settings.

As thus calving is determining current shape, it unclear how much additional warming is needed before the ablation becomes strong enough to push the ice sheet from the shore. As the authors show in figure 9, this happens first and only along the southwestern margin, but never in the North. One may doubt if the results of Fyke 2011 are the most realistic LIG retreat evolution presented so far, but a GIS that firmly stick to the northern shore of Greenland throughout the whole LIG is a clear indication that the presented model isn't providing a realistic evolution estimate.

I know, the authors discuss this on p12, but that makes the results not trustworthier.

**Other (major) comments**

*Section 3.1*

- As also commented at figure 2 (see below), the authors are complaining about something that can be resolved. Correct NorESM temperatures for the topographic differences and now one can discuss whether SICOPOLUS get correct temperatures or not. My feeling: NorESM is too cold, but how much colder was

PI compared to now. By the way, why using ERA40 if we have ERA-Interim? And which period of ERA40 is used?

- What actually done for downscaled (=20 km) land points outside the NorESM land domain?

- The authors entirely neglect to evaluate runoff – in this study modeled melt – and hence the SMB. Add the modeled downscaled SMB for the present day topography (using default PDD/ITM settings) in Figure 2 and compare this with an estimate from MAR, RACMO or another state-of-the-art model/observational estimate and discuss, including an estimation of the PI-to-now difference of the GIS SMB.

- Also for the LIG slides, add downscaled modeled SMB using the present day topography. It is maybe better to show for each time slice the downscaled SMB instead of the difference compared to the downscaled SMB for PI. After this improvement the reader has a clue what to expect from the LIG forcing on the GIS (P9 L27-28).

*Other sections*

P4 L2: Which specific CESM version was branched for NorESM?

P10 L15: As the GIS reaches the coast where it shouldn't it is not (only) due to insufficient calving, but also due to insufficient melting. And for the latter the authors can't excuse by saying "it's SIA".

S3.3: State that this will be discussed in S4.3. The section raises comments which are partly addressed there.

S3.4: See above

[Figure]

S4.1.1 Comparing surface temperatures is senseless if topographies are very different. It is not stated in the manuscript if such a correction is applied; therefore, it is hard to compare various methods.

Besides, the section does not – at least for me – provide a clear conclusion what the analysis say about the results presented in this manuscript.

S4.1.3: The authors suggest that the only method to feed back the shrinking ice sheet into the GCM is through fully coupled simulations. There is, of course, a middle road, namely GCM time slice runs using update topography (e.g. Helsen et al, 2013 did using a RCM).

S4.3: Besides that the GIS extended well up to continental shelf, it also connected with the Laurentide Ice Sheet across Nares Strait. LIG ice sheet presented here is thus likely too small.

What remains unclear: how does this affect the results presented?

P16 L15: As we are quite certain that the preLIG GIS extended on the continental shelf and the LIG GIS extend is also quite bound by sea-level rise proxies and ice core data, the logical conclusion is that it melted thus faster during LIG than the melt rates now required to get to a logical LIG minimum from a too small preLIG GIS. The authors should doubt what they tune (melt rate) not to what they tune to (GIS extent).

F2: This figure must be improved substantially. In the current version it is not clear how the land/sea distribution in NorESM was and the comparison to ERA40 is obstructed by the elevation difference. Furthermore, which temperature is shown? Is it 2m temperature or surface temperature? My suggestion: Left row: use NorESM land/sea mask and remove interpolation (ncl: @cnFillMode = "Raster-Fill"). Right row: difference with ERA40 after correction for elevation difference

using the default lapse rate. The authors now can use a higher-resolution land-sea mask as is done now.

F3: Again, a figure that should have been much clearer. I think, however, that this plot is not needed after all. The authors should add elevation contours to figure 2 (NorESM elevation in the left row, true elevation in the right row) and remove this figure. Furthermore, include the ice sheet outline in NorESM if different from the land/sea mask.

F4: again, use the NorESM land/sea mask and include the NorESM topography and ice sheet extend.

F5: In the case that this figure stays, it should be improved. The figure is unclear on the result for varying PDD factor combinations. Solved this by using (exclusively) red, blue and green for the three PDD snow factors and a horizontal displacement (as in the legend) to distinguish the different PDD ice factors.

F12, F13: Temperature differences are meaningless unless it is clear what the elevation difference is.

---

## Referee Comment (RC2) · Anonymous Referee #2 · 14 Apr 2016

Understanding the stability of the Greenland Ice Sheet (GrIS) during higher than present temperatures is essential, in particular for deriving better projections of future ice sheet mass loss. The reconstruction of the extent and volume of the GrIS during the Last Interglacial period (LIG) has remained a challenge. Significant differences in published results are attributed to differences in applied climate forcing and the methods for coupling the ice sheet and the climate models.

This study adds to the growing literature of ice sheet melt and sea-level contribution reconstructions from Greenland Ice Sheet melt during the LIG. However, it does not

offer an improvement over previous estimates because of the limitations of the applied methodology. The approach employs SICOPOLIS, a thermomechanically coupled ice sheet model using the Shallow Ice Approximation (SIA) and the Norwegian Earth System Model (NorESM) without fully coupling the two components. Ice sheet melt is estimated using PDD modeling with temperatures adjusted to elevation changes by using spatially and temporarily constant lapse rates.

Although widely used for investigating ice sheet evolution in millennial to multi-million year time scales, the approach has several caveats. Most importantly, PDD factors of snow and ice are not constant in space and time and therefore the PDD method has a poor performance for reconstructing of melt-rate histories. Instead of the PDD method, the authors should have used one of the more realistic and accurate approaches. Such methods do exist, including the insolation-temperature-melt model from Robinson et al., (2011) or the SMB gradient method, introduced by Helsen et al. (Helsen et al., 2012, Coupling of climate models and ice sheet models by surface mass balance gradients: application to the Greenland Ice Sheet. The Cryosphere, 6(2), 255–272. http://doi.org/10.5194/tc-6-255-2012).

Another main limitation of the study is that rather than evolving the ice sheet surface with the changing climate, the present-day ice sheet topography is used for the climate simulation. Relatively high elevations during the LIG likely cause an underestimation of the warming in the LIG, which could result in low estimates of ice loss. The authors dismiss the potential for performing a fully coupled model experiment as computationally too expensive at the present. However, this challenge is no longer prohibitive, as demonstrated by the first coupled ice sheet/climate model, which is currently in discussion in Past Climate (Goelzer, H., Huybrechts, P., Loutre, M. F., & Fichefet, T. (2016). Last Interglacial climate and sea-level evolution from a coupled ice sheet-climate model. Climate of the Past Discussions, 0, 1–34. http://doi.org/10.5194/cp-2015-175).

The authors argue that by using present-day observations and paleo ice core reconstructions to evaluate a range of model results corresponding to different parameter selection (basal sliding, PDD factors and lapse rate) they could obtain a robust estimate of LIG GrIS lost, despite the significant model simplifications. However, this appears to be contradicted by the fact that using the same methodology with similar parameterization but a different climate model that features warmer sea and air temperatures in N Greenland, Born and Nisancioglu (2012) have obtained a completely different pattern of LIG mass loss during the LIG with a strong ice sheet retreat in NE Greenland.

Detailed remarks: Page 2, lines 15-19: the ice penetrating radar mapping only detect the ice that is still present. However, Eemian ice could have thinned and melted in southern Greenland and therefore no longer present.

Page 5, lines 3-9: The modification of the global heatflux estimate from Pollack et al., 1993 by Greve (2005) results in an improved match of basal temperatures measured in the boreholes, but still provides an unrealistic distribution of geothermal heatflow. Rogozhina et al. (Rogozhina, I., Hagedoorn, J. M., Martinec, Z., Fleming, K., Soucek, O., Greve, R., & Thomas, M. (2012). Effects of uncertainties in the geothermal heat flux distribution on the Greenland Ice Sheet: An assessment of existing heat flow models. Journal of Geophysical Research, 117(F2), F02025. http://doi.org/10.1029/2011JF002098) have shown a spatially uniform geothermal heatflow distribution provides the best fit to the borehole temperature observations.

Page 5, lines 12-22: using the techniques mentioned above (ITM or SMB gradient) instead of PDD would eliminate the need of relying on the traditional lapse rate assumptions, thus improve the estimate of surface melt

Page 5, lines 22-25: what is the source of this particular parameterization – please include reference

Page 7, lines 7-12: the motivation of varying the basal sliding parameter is not clear. I assume that the defaults SICOPOLIS value was selected to provide a good match

for the PI GrIS. The basal sliding parameter, which is usually inferred by parameter assimilation, varies spatially and temporarily. It is not clear what improvement is offered by employing different, constant basal sliding parameters.

MacGregor et al., 2015 reference is missing

Figure 2: why not using ERA-interim or an observation-based reconstruction (e.g., Bales, R. C., McConnell, J. R., Mosley-Thompson, E., & Csatho, B. (2001). Accumulation over the Greenland ice sheet from historical and recent records. Journal of Geophysical Research: Atmospheres (1984–2012), 106(D24), 33813–33825. http://doi.org/10.1029/2001JD900153)

Figure 3: why is the surface elevation in NorESM-L compared to ETOPO1? ETOPO is not used in the study – should it be Bamber et al., 2013?

Figure 5: Units of PDD factors should be added

[Figure]

---

## Author Comment (AC1) · 4 May 2016

**Reply to comments of Referee #1**

We thank referee #1 for an extensive and constructive review. Below we discuss all concerns and questions, and give a perspective of how we will address these in a revised manuscript. A more in-depth reply to the comments will be provided with the revised manuscript.

*Referee #1:*
*The authors present a study of the evolution of the Greenland Ice Sheet (GIS) during the Last Interglacial (LIG), using time slices from the GCM NorESM and ice model SICOPOLIS and select the most suitable model outcome using extend and local elevation constraints from ice core data. I firstly make my apologies for the long time I've needed to complete this review, and secondly make my apologies that I'm not very positive concerning the research presented. The authors use the PDD (positive degree day) method, which is not applicable here, and the results present never look that realistic that it earns some reliability. The latter (an ice sheet glued to the Arctic Ocean) could be bad luck, but using the PDD is a method flaw that could have been avoided. To my opinion the results presented in this manuscript does not provide any new information on the LIG climate, LIG GIS extend or the GIS sensitivity to climate change, nor in how we could tackle this problem. In the cryospheric community, we are generally kind, but by using a method (PDD) that is unsuitable, it leaves me with no other option that to advise the editor to reject this manuscript unless the authors redo the ice model simulations using an ITM method to derive melt rates from the GCM output.*

*Referee #1: Experiment setup: PDD*
*As the authors know, the PDD is the traditional "way out" if someone has no surface energy balance (SEB) data to estimate melt and runoff. It does work after careful tuning – yes I know - if cloud and insolation characteristics remain similar. Hence, it works for each glacier individually for time scales up to a few centuries. For multi-millennia simulations over the entire GIS the assumption of two constant PDD factors for snow and ice does not hold, neither spatially nor in time. Bluntly said, as PDD is not valid here, it should not have been used. Period. And as melt is the key process that drives the evolution of the GIS during the LIG, it's lethal for this manuscript. The (for me obvious) method to use is the an Insolation-Temperature Method (ITM, e.q. Robison et al, 2011, Climate of the Past) which explicitly includes insolation into the derivation of melt/runoff. And also an ITM allows tuning until the model results start to resemble past and current GIS states. Honestly, I can't understand why the authors did not take the effort to implement this. It is not complicated, neither state of the art. A state of the art approach would be a method like presented in Vizcaino and others, J.Clim., 2013. There is no longer an excuse for using PDD.*

Reply:
First of all we would like to note that the PDD method is still widely applied in paleoclimate research. The main reason for this is its simplicity. Yes, we agree that the parameterization is very simple, but that also prevents over-tuning of the climate forcing. As we show in the manuscript, even for recent past warm time periods such as the last interglacial, the climates models simulate very different climates. High resolution regional climate models will give better spatial resolution, but these are only as good as the global models that force them.

We appreciate the reviewer's suggestion of using the ITM model. This will include an additional factor which is direct related to insolation. However, the added value of including insolation as a direct forcing for surface melt also has its challenges due to the added uncertainty of the transmissivity of the atmosphere (an essential parameter in the ITM scheme). This transmissivity is taken as a linear function of elevation (Robinson et al., 2010), but this again (like in the PDD scheme) assumes that cloud characteristics remain constant over time. Another method, as also referred to by the second referee, is the SMB gradient method introduced by Helsen et al. (2012) where the SMB is adjusted locally depending on elevation changes. The necessary SMB gradients

are first derived from a regional climate model, applying linear regressions to SMB and elevation data within a small zone surrounding each grid point. This approach is quite sensitive to uncertainties in elevation, where an elevation uncertainty of $\pm 100$ m translates to an uncertainty of 1.2 m on the simulated ~2.1 m of last interglacial maximum sea level rise, dominating the total uncertainty (Helsen et al., 2013).

An approach where the ice sheet model and its surface mass balance are fully integrated in an Earth System Model (like CESM in Vizcaino et al., 2013) is impossible to run on the timescales investigated there (more than 20,000 years - not counting the spin-up period). The first fully coupled ice sheet climate model simulating over the entire last interglacial is currently in discussion in Climate of the Past (Goelzer et al.). While this is a great technical effort, this study also applied the PDD scheme to calculate the melt over the ice sheets. Goelzer et al. use an ESM of intermediate complexity with a spatial resolution of only T21 (approximately 5-6°) and 3 levels in the atmosphere, which indicates the limitations of fully coupled ESM-ice sheet models for paleo simulations.

We realize that our results are likely to depend on the melt scheme applied. As suggested by the reviewer, we will investigate the impact of adding the direct impact of insolation in the ITM scheme and will adjust our manuscript accordingly.

*Referee #1: Model performance of PI & LIG climate*
*Figure 6a and b does not give a very affirming feeling that the NorESM & SICOPOLIS combination can realistically model the GIS. The figures show that the model let the pre-industrial (PI) ice sheet grow until it meets the shore. (As SICOPOLIS does not model ice shelves, it can't grow into the sea but if it could, it even might have grown on the continental shelf). Figure 6ab says me that melt and runoff are largely underestimated for the default settings.*
*As thus calving is determining current shape, it unclear how much additional warming is needed before the ablation becomes strong enough to push the ice sheet from the shore. As the authors show in figure 9, this happens first and only along the south- western margin, but never in the North. One may doubt if the results of Fyke 2011 are the most realistic LIG retreat evolution presented so far, but a GIS that firmly stick to the northern shore of Greenland throughout the whole LIG is a clear indication that the presented model isn't providing a realistic evolution estimate.*
*I know, the authors discuss this on p12, but that makes the results not trustworthier.*
Reply:
Correct, the melt and runoff are underestimated in the default set-up. This is maybe more clearly shown in Figure 7 where we compare the SMB to the total accumulation over Greenland. This is one of the reasons why we find that higher PDD factors are more appropriate. We will explain this more clearly in the revised manuscript and update Figure 2 showing our best PI simulation, instead of our default simulation.

Note, however, that even for the best fit PI simulations, ice extents to most of the Northern coast. This is thought to be a result of the Arctic cold bias in the climate forcing simulated by NorESM. Note also that there is a large range in simulated Arctic climates as given by state of the art climate models (e.g. IPCC 2013, Ch9). The NorESM bias is also causing the persistence of ice in northern Greenland during the LIG simulations. PI July temperatures from NorESM are up 2.5°C below freezing for the Northern coast (Fig. 2), and the LIG maximum warming is simulated to be 1-2°C (Fig. 4), which is not sufficient to cause large scale ablation.

We realize that the existence of ice in northern Greenland during the LIG is disputed, but it does not effect the main conclusions of the manuscript (~2m sea level rise during LIG between 124 and 122

ka). In order to better constrain the very uncertain past retreat of ice on northern Greenland we would need ice core or geological data, which is currently not available.

Note however, that by including a direct impact of insolation, as in the ITM method, it is possible that the change in the gradient of insolation during the LIG could give a slight increase in melt over northern Greenland. This is a point we will investigate in the revised manuscript.

*Referee #1: Section 3.1*
*- As also commented at figure 2 (see below), the authors are complaining about something that can be resolved. Correct NorESM temperatures for the topographic differences and now one can discuss whether SICOPOLUS get correct temperatures or not. My feeling: NorESM is too cold, but how much colder was PI compared to now.*

Reply: see also reply to comments on Figure 2 and 3. When adjusting for the topographic differences one needs to choose a temperature lapse rate. We will update the figures (as stated below) and will give an indication of the model bias compared to ERA40, depending on the different lapse rate corrections.

*Referee #1: By the way, why using ERA40 if we have ERA-Interim? And which period of ERA40 is used?*

Reply: We choose to compare to the mean values of the entire period of ERA40, and not ERA-Interim, because 1) ERA-Interim has a shorter number of years included, and 2) ERA-Interim covers more recent year, including some with strong warming. We compare these data products to a pre-industrial climate model simulation that does not cover the current climate warming, and is in an equilibrium state. We will better explain this in the revised manuscript.

*Referee #1: What actually done for downscaled (=20 km) land points outside the NorESM land domain?*

Reply: NorESM works not only with a land/sea mask, but every gridbox has a fractional area corresponding to land. Because of this there is no downscaled land point that is entirely ocean in NorESM. See figure LANDFRAC_NorESM.png below:

Fraction of sfc area covered by land

*Referee #1: The authors entirely neglect to evaluate runoff – in this study modeled melt – and hence the SMB. Add the modeled downscaled SMB for the present day topography (using default PDD/ITM settings) in Figure 2 and compare this with an estimate from MAR, RACMO or another state-of-the-art model/observational estimate and discuss, including an estimation of the PI-to-now difference of the GIS SMB.*

Indeed we did not show the spatial pattern of the simulated SMB; rather we show the mass balance partitioning following Robinson et al. (2011) in Figure 7. However, in the revised manuscript we will add and compare the SMB patterns for the different simulations to a model/observational estimate. As an example we included here the SMB of the best PI simulation compared to RACMO (1960-2010; Van Angelen et al, 2013).

[Figure]

*Referee #1: Also for the LIG slides, add downscaled modeled SMB using the present day topography. It is maybe better to show for each time slice the downscaled SMB instead of the difference compared to the downscaled SMB for PI. After this improvement the reader has a clue what to expect from the LIG forcing on the GIS (P9 L27-28).*
Reply: Thanks, we will include figures showing the LIG SMB in the revised manuscript.

*Referee #1: P4 L2: Which specific CESM version was branched for NorESM?*
Reply: The following NorESM components are based on the CESM: atmospheric component CAM4, land component CLM4, and sea ice component CICE4. The ocean model used in NorESM is based on MICOM, but largely modified. For more information we refer to Zhang et al., 2012.

*Referee #1: P10 L15: As the GIS reaches the coast where it shouldn't it is not (only) due to insufficient calving, but also due to insufficient melting. And for the latter the authors can't excuse by saying "it's SIA".*
Reply: Thanks, we will rewrite this in the revised manuscript.

*Referee #1: S3.3: State that this will be discussed in S4.3. The section raises comments which are partly addressed there.*
*S3.4: See above*
Reply: Thanks, this will be done.

*Referee #1: S4.1.1: Comparing surface temperatures is senseless if topographies are very different. It is not stated in the manuscript if such a correction is applied; therefore, it is hard to compare various methods. Besides, the section does not – at least for me – provide a clear conclusion what the analysis say about the results presented in this manuscript.*
Reply: No correction for the topographies is applied. For the LIG simulations this has no effect as the difference to PI temperatures is discussed, and all these model simulations do not adapt their Greenland ice sheet topography for the LIG. For the PI comparison this could be an important effect, which we will assess in the revised version of the manuscript.

*Referee #1: S4.1.3: The authors suggest that the only method to feed back the shrinking ice sheet into the GCM is through fully coupled simulations. There is, of course, a middle road, namely GCM time slice runs using update topography (e.g. Helsen et al, 2013 did using a RCM).*
Reply: That is correct, we will revise as appropriate. Please note that while Helsen et al. (2013) couple their RCM and ice sheet model every 1.5 kyrs, they only use 2 GCM time slice simulations as boundaries for their forcing (at 125 and 115ka). In contrast we do not have a RCM, but do update the GCM climate forcing every 5 kyrs (at 130, 125, 120 and 115ka).

*Referee #1: S4.3: Besides that the GIS extended well up to continental shelf, it also connected with the Laurentide Ice Sheet across Nares Strait. LIG ice sheet presented here is thus likely too small. What remains unclear: how does this affect the results presented?*
Reply: We suggest that the LIG GIS simulation should start with a larger than PD ice sheet (in contrast to some previous studies). However, until more data is available on the exact glacial extent, investigating the possible connection of the GIS with the Laurentide is highly uncertain. If time permits we will consider running multiple spin-up experiments. However, we are of the opinion that there are other factors that should be investigated first.

*Referee #1: P16 L15: As we are quite certain that the preLIG GIS extended on the continental shelf and the LIG GIS extend is also quite bound by sea-level rise proxies and ice core data, the logical conclusion is that it melted thus faster during LIG than the melt rates now required to get to a logical LIG minimum from a too small preLIG GIS. The authors should doubt what they tune (melt rate) not to what they tune to (GIS extent).*
Reply: Agree, these sentences will be adjusted in a revised manuscript.

*Referee #1: F2: This figure must be improved substantially. In the current version it is not clear how the land/sea distribution in NorESM was and the comparison to ERA40 is obstructed by the elevation difference. Furthermore, which temperature is shown? Is it 2m temperature or surface temperature? My suggestion: Left row: use NorESM land/sea mask and remove interpolation (ncl: @cnFillMode = "Raster- Fill"). Right row: difference with ERA40 after correction for elevation difference using the default lapse rate. The authors now can use a higher-resolution land-sea mask as is done now.*
*F3: Again, a figure that should have been much clearer. I think, however, that this plot is not needed after all. The authors should add elevation contours to figure 2 (NorESM elevation in the left row, true elevation in the right row) and remove this figure. Furthermore, include the ice sheet outline in NorESM if different from the land/sea mask.*
Reply: We combine our reply to these two comments, as they are connected. By showing all maps in the SICOPOLIS interpolated grid we intended to make it easy to compare the different figures in the manuscript. However, you are right that in this way we cannot show the basic input (NorESM)

precisely. We will adjust the left row of Figure 2 to show the raw NorESM fields. The reason for not showing the elevation corrected difference between NorESM and ERA40 is that the temperature lapse rate is not well defined (something we test in our sensitivity simulations). However, it will make the comparison easier for the reader if we assume a constant lapse rate (e.g. 6.5degC/km), and adjust the figures to follow the referee's suggestions. As an addition we will discuss the effects of a higher and lower lapse rate.

We show the surface temperatures as simulated by the atmospheric component of NorESM.

*Referee #1: F4: again, use the NorESM land/sea mask and include the NorESM topography and ice sheet extend.*

Reply: We can plot these on the original NorESM grid and show the land/sea mask and ice sheet extent. But adding the NorESM topography here does not help, because these difference plots are not affected by the topography (which is kept constant).

*Referee #1: F5: In the case that this figure stays, it should be improved. The figure is unclear on the result for varying PDD factor combinations. Solved this by using (exclusively) red, blue and green for the three PDD snow factors and a horizontal displacement (as in the legend) to distinguish the different PDD ice factors.*

Reply: Thanks, we will improve the figure accordingly, if it stays in the manuscript after discussing the different melt schemes.

*Referee #1: F12, F13: Temperature differences are meaningless unless it is clear what the elevation difference is.*

Reply: see comments above.

**References:**

Flato, G., J. Marotzke, B. Abiodun, P. Braconnot, S.C. Chou, W. Collins, P. Cox, F. Driouech, S. Emori, V. Eyring, C. Forest, P. Gleckler, E. Guilyardi, C. Jakob, V. Kattsov, C. Reason and M. Rummukainen, 2013: Evaluation of Climate Models. In: *Climate Change 2013: The Physical Science Basis. Contribution of Working Group I to the Fifth Assess- ment Report of the Intergovernmental Panel on Climate Change* [Stocker, T.F., D. Qin, G.-K. Plattner, M. Tignor, S.K. Allen, J. Boschung, A. Nauels, Y. Xia, V. Bex and P.M. Midgley (eds.)]. Cambridge University Press, Cambridge, United Kingdom and New York, NY, USA.

Van Angelen, J. H., M. R. van den Broeke, B. Wouters and J. T. M. Lenaerts, 2013: Contemporary (1960–2012) evolution of the climate and surface mass balance of the Greenland ice sheet, Surveys of Geophysics, doi: 10.1007/s10712-013-9261-z.

Zhang, Z. S., Nisancioglu, K., Bentsen, M., Tjiputra, J., Bethke, I., Yan, Q., Risebrobakken, B., Andersson, C., and Jansen, E.: Pre-industrial and mid-Pliocene simulations with NorESM-L, Geosci. Model Dev., 5, 523–533, doi:10.5194/gmd-5-523-2012, 2012.

---

## Author Comment (AC2) · 4 May 2016

**Reply to comments of Referee #2**

We thank referee #2 for an extensive and constructive review. Below we discuss all concerns and questions, and give a perspective of how we will address these in a revised manuscript. A more in-depth reply to the comments will be provided with the revised manuscript.

*Referee #2: Understanding the stability of the Greenland Ice Sheet (GrIS) during higher than present temperatures is essential, in particular for deriving better projections of future ice sheet mass loss. The reconstruction of the extent and volume of the GrIS during the Last Interglacial period (LIG) has remained a challenge. Significant differences in published results are attributed to differences in applied climate forcing and the methods for coupling the ice sheet and the climate models.*
*This study adds to the growing literature of ice sheet melt and sea-level contribution reconstructions from Greenland Ice Sheet melt during the LIG. However, it does not offer an improvement over previous estimates because of the limitations of the applied methodology. The approach employs SICOPOLIS, a thermomechanically coupled ice sheet model using the Shallow Ice Approximation (SIA) and the Norwegian Earth System Model (NorESM) without fully coupling the two components. Ice sheet melt is estimated using PDD modeling with temperatures adjusted to elevation changes by using spatially and temporarily constant lapse rates.*

*Referee #2: Although widely used for investigating ice sheet evolution in millennial to multi-million year time scales, the approach has several caveats. Most importantly, PDD factors of snow and ice are not constant in space and time and therefore the PDD method has a poor performance for reconstructing of melt-rate histories. Instead of the PDD method, the authors should have used one of the more realistic and accurate approaches. Such methods do exist, including the insolation-temperature-melt model from Robinson et al., (2011) or the SMB gradient method, introduced by Helsen et al. (Helsen et al., 2012, Coupling of climate models and ice sheet models by surface mass balance gradients: application to the Greenland Ice Sheet. The Cryosphere, 6(2), 255–272. http://doi.org/10.5194/tc-6-255-2012).*
Reply:
Indeed, the PDD method is widely applied in paleoclimate research. The main reason for this is its simplicity. Yes, we agree that the parameterization is very simple, but that also prevents over-tuning of the climate forcing. As we show in the manuscript, even for recent past warm time periods such as the last interglacial, the climates models simulate very different climates. High resolution regional climate models will give better spatial resolution, but these are only as good as the global models that force them.

We appreciate the reviewer's suggestion of using alternative approaches. The first suggestion, the ITM model, includes an additional factor which is direct related to insolation. However, the added value of including insolation as a direct forcing for surface melt also has its challenges due to the added uncertainty of the transmissivity of the atmosphere (an essential parameter in the ITM scheme). This transmissivity is taken as a linear function of elevation (Robinson et al., 2010), but this again (like in the PDD scheme) assumes that cloud characteristics remain constant over time. In the second method, the SMB gradient method of Helsen et al. (2012), the SMB is adjusted locally depending on elevation changes. The necessary SMB gradients are first derived from a regional climate model, applying linear regressions to SMB and elevation data within a small zone surrounding each grid point. This approach is quite sensitive to uncertainties in elevation, where an elevation uncertainty of $\pm$100 m translates to an uncertainty of 1.2 m on the simulated ~2.1 m of last interglacial maximum sea level rise, dominating the total uncertainty (Helsen et al., 2013).

We realize that our results are likely to depend on the melt scheme applied. As suggested by the reviewers, we will investigate the impact of the different schemes and will adjust our manuscript accordingly.

*Referee #2: Another main limitation of the study is that rather than evolving the ice sheet surface with the changing climate, the present-day ice sheet topography is used for the climate simulation. Relatively high elevations during the LIG likely cause an underestimation of the warming in the LIG, which could result in low estimates of ice loss. The authors dismiss the potential for performing a fully coupled model experiment as computationally too expensive at the present. However, this challenge is no longer prohibitive, as demonstrated by the first coupled ice sheet/climate model, which is currently in discussion in Past Climate (Goelzer, H., Huybrechts, P., Loutre, M. F., & Fichefet, T. (2016). Last Interglacial climate and sea-level evolution from a coupled ice sheet- climate model. Climate of the Past Discussions, 0, 1–34. http://doi.org/10.5194/cp- 2015-175).*

Reply: While the work of Goelzer et al. presents a great technical effort, its results are limited by the relatively low resolution of the Earth System Model used (3 levels of approximately 5-6° spatial resolution in the atmosphere, compared to 26 levels and 3-4° spatial resolution in NorESM-L), the intermediate complexity model LOVECLIM, requiring an even larger amount of downscaling/interpolation of the simulated climate fields. Furthermore this study also applies the PDD method to simulate melt over the ice sheets. As discussed on page 15, lines 1-5, we refer to the study of Stone et al. (2013), who use a similar set-up as we do, but additionally tested how a smaller GIS affects the climate simulated by their GCM. It is unlikely that the GIS became ice-free during the LIG, and a small reduction of the ice sheet does not have a large effect on the large-scale climate as simulated by the GCMs.
We will further clarify this in the revised manuscript.

*Referee #2: The authors argue that by using present-day observations and paleo ice core reconstructions to evaluate a range of model results corresponding to different parameter selection (basal sliding, PDD factors and lapse rate) they could obtain a robust estimate of LIG GrIS lost, despite the significant model simplifications. However, this appears to be contradicted by the fact that using the same methodology with similar parameterization but a different climate model that features warmer sea and air temperatures in N Greenland, Born and Nisancioglu (2012) have obtained a completely different pattern of LIG mass loss during the LIG with a strong ice sheet retreat in NE Greenland.*

Reply: There are several large differences between the study of Born and Nisancioglu (2012), hereafter BN2012, and the one described here. Most important differences are: 1) Indeed the climate forcing was different (different models were used, as well as only 2 time slices), which has a major impact; 2) BN2012 initialize their LIG simulations with present-day ice and bedrock topographies; and 3) different data was used to constrain the model simulations. The combined effect of this is that BN2012 simulate a higher sea level contribution (see Figure 1). We will more clearly discuss the difference in the revised manuscript.

*Referee #2: Detailed remarks: Page 2, lines 15-19: the ice penetrating radar mapping only detect the ice that is still present. However, Eemian ice could have thinned and melted in southern Greenland and therefore no longer present.*

Reply: Correct, we will rewrite.

*Referee #2: Page 5, lines 3-9: The modification of the global heatflux estimate from Pollack et al., 1993 by Greve (2005) results in an improved match of basal temperatures measured in the boreholes, but still provides an unrealistic distribution of geothermal heatflow. Rogozhina et al. (Rogozhina, I., Hagedoorn, J. M., Martinec, Z., Fleming, K., Soucek, O., Greve, R., & Thomas, M. (2012). Effects of uncertainties in the geothermal heat flux distribution on the Greenland Ice Sheet:*

*An assessment of existing heat flow models. Journal of Geophysical Research, 117(F2), F02025. http://doi.org/10.1029/2011JF002098) have shown a spatially uniform geothermal heatflow distribution provides the best fit to the borehole temperature observations.*
Reply: Thank you, we were not aware of this study. We will test what effect a different heatflux pattern has on our results and include in the revised manuscript.

*Referee #2: Page 5, lines 12-22: using the techniques mentioned above (ITM or SMB gradient) instead of PDD would eliminate the need of relying on the traditional lapse rate assumptions, thus improve the estimate of surface melt*
Reply: The ITM method still relies on surface temperatures, but indeed for the SMB gradient method we could eliminate the need of the temperature lapse rate. Please see our discussion above concerning the implementation of the different methods. We will test the impact of using ITM in the revised manuscript.

*Referee #2: Page 5, lines 22-25: what is the source of this particular parameterization – please include reference*
Reply: This default parameterization in SICOPOLIS is based on Marsiat (1994); a reference will be included.

*Referee #2: Page 7, lines 7-12: the motivation of varying the basal sliding parameter is not clear. I assume that the defaults SICOPOLIS value was selected to provide a good match for the PI GrIS. The basal sliding parameter, which is usually inferred by parameter assimilation, varies spatially and temporarily. It is not clear what improvement is offered by employing different, constant basal sliding parameters.*
Reply: The basal sliding parameter is a tuned constant parameter in SICOPOLIS that relates the basal sliding velocity to the basal shear stress. This is commonly taken as a constant value, as the distribution of sediments and bedrock irregularities is not well known for the entire ice sheet. Parameter assimilation or inversion could give a spatially varying parameter, but without indications how this will vary over time. In order to avoid unknown details for longer simulations the parameter is kept constant. However, we assess the effect this constant value has on our ice sheet results. We will better clarify this in the revised manuscript.

*Referee #2: MacGregor et al., 2015 reference is missing*
Reply: We will add this.

*Referee #2: Figure 2: why not using ERA-interim or an observation-based reconstruction (e.g., Bales, R. C., McConnell, J. R., Mosley-Thompson, E., & Csatho, B. (2001). Accumulation over the Greenland ice sheet from historical and recent records. Journal of Geophysical Research: Atmospheres (1984–2012), 106(D24), 33813–33825. http://doi.org/10.1029/2001JD900153)*
Reply: We prefer to use the 40 years long reconstruction of ERA40, because we compare this data product to a pre-industrial climate simulation that does not cover the current climate warming, and is in an equilibrium state. We will better explain this in the revised manuscript.

*Referee #2: Figure 3: why is the surface elevation in NorESM-L compared to ETOPO1? ETOPO is not used in the study – should it be Bamber et al., 2013?*
Reply: On this scale (20 km grid of SICOPOLIS) the ETOPO1 and Bamber et al. (2013) topographies are very similar. However, to be more consistent we will adjust the comparison to use the Bamber topography in the revised manuscript.

*Referee #2: Figure 5: Units of PDD factors should be added*
Reply: We will include the units.

**References:**

MacGregor, J. A., M. A. Fahnestock, G. A. Catania, J. D. Paden, S. P. Gogineni, S. K. Young, S. C. Rybarski, A. N. Mabrey, B. M. Wagman, and M. Morlighem (2015), Radiostratigraphy and age structure of the Greenland Ice Sheet, J. Geophys. Res. Earth Surf., 120, 212–241, doi:10.1002/2014JF003215.

Marsiat, I. 1994. Simulation of the northern hemisphere continental ice sheets over the last glacial–interglacial cycle: experiments with a latitude–longitude vertically integrated ice sheet model coupled to zonally averaged climate model. *Paleoclimates*, **1**, 59–98.